# Implementation of an antibody characterization procedure and application to the major ALS/FTD disease gene C9ORF72

Carl Laflamme[1], Paul M McKeever[2,3], Rahul Kumar[1], Julie Schwartz[1], Mahshad Kolahdouzan[4], Carol X Chen[1], Zhipeng You[1], Faiza Benaliouad[1], Opher Gileadi[5], Heidi M McBride[1], Thomas M Durcan[1], Aled M Edwards[1,6], Luke M Healy[4], Janice Robertson[2,3], Peter S McPherson[1]*

[1]Tanenbaum Open Science Institute, Montreal Neurological Institute, McGill University, Montreal, Canada; [2]Tanz Centre for Research in Neurodegenerative Diseases, University of Toronto, Toronto, Canada; [3]Department of Laboratory Medicine and Pathobiology, University of Toronto, Toronto, Canada; [4]Neuroimmunology Unit, Montreal Neurological Institute, McGill University, Montreal, Canada; [5]Structural Genomics Consortium, University of Toronto, Toronto, Canada; [6]Structural Genomics Consortium, Nuffield Department of Clinical Medicine, University of Oxford, Oxford, United Kingdom

**Abstract** Antibodies are a key resource in biomedical research yet there are no community-accepted standards to rigorously characterize their quality. Here we develop a procedure to validate pre-existing antibodies. Human cell lines with high expression of a target, determined through a proteomics database, are modified with CRISPR/Cas9 to knockout (KO) the corresponding gene. Commercial antibodies against the target are purchased and tested by immunoblot comparing parental and KO. Validated antibodies are used to definitively identify the most highly expressing cell lines, new KOs are generated if needed, and the lines are screened by immunoprecipitation and immunofluorescence. Selected antibodies are used for more intensive procedures such as immunohistochemistry. The pipeline is easy to implement and scalable. Application to the major ALS disease gene *C9ORF72* identified high-quality antibodies revealing C9ORF72 localization to phagosomes/lysosomes. Antibodies that do not recognize C9ORF72 have been used in highly cited papers, raising concern over previously reported C9ORF72 properties.

*For correspondence:
peter.mcpherson@mcgill.ca

**Competing interests:** The authors declare that no competing interests exist.

## Introduction

Antibodies are vital tools in the biomedical research arsenal and in keeping with their importance there are over a million unique antibodies that are commercially available (Dr. Matt Baker, Thermo-Fisher, personal communication). In addition are the outputs of numerous, publicly-funded projects, which have generated recombinant or monoclonal antibodies, thus creating renewable antibodies with the potential for high reproducibility (*Hornsby et al., 2015*; *Marcon et al., 2015*; *Na et al., 2016*; *Venkataraman et al., 2018*; *Andrews et al., 2019*). The largest public antibody initiative has generated greater than 24,000 antibodies corresponding to almost 17,000 human genes, with a focus on polyclonal antibodies (*Uhlén et al., 2015*). Thus, there are pre-existing antibodies for a large percentage of the human genome.

The pace and attention given to antibody generation has not been matched with an equal effort on antibody characterization or in implementing standardized antibody characterization procedures.

As a result, most products are generated and then sold/distributed with only rudimentary and non-quantitative supporting data, and as a result there are serious flaws in the reliability of many of the available reagents. Ill-defined antibodies contribute significantly to a lack of reproducibility in important research efforts including preclinical studies, with estimates that up to 90% of a select group of 53 landmark preclinical studies suffered from such flaws (*Bradbury and Plückthun, 2015*).

Several *ad hoc* international working groups have met to help define best practices for antibody validation (*Taussig et al., 2018*). One of the groups (*Uhlen et al., 2016*) proposed five separate validation criteria: (1) genetic strategies in which the specificity of the antibody toward the endogenous protein is confirmed by the loss of signal in cells or tissues comparing parental to knockout (KO) or knockdown controls; (2) orthogonal strategies in which correlations are made between the antibody signal and known information regarding protein abundance or localization; (3) the ability of two independent uncharacterized antibodies recognizing different epitopes in the same target protein to recognize the same protein; (4) using overexpressed epitope-tagged proteins comparing antibodies against the tag to the uncharacterized antibody; (5) immunoprecipitation followed by mass spectrometry to determine if the protein of interest is a major signal in the sample. These criteria are arguably not of equal scientific value and there is no consensus as to which should be used. The first and fifth methods are the most unbiased and useful, whereas the remaining are less informative and perhaps flawed. The genetic approaches presented in criterion one are suitable for antibody validation in all applications, yet there is no template for their application.

Historically, the lack of a suitable control – an isogenic source of proteins lacking the target antigen, has hampered the implementation of criterion 1, but this has changed: it is now routine to make KO cell lines in an array of cell types which, for non-essential proteins, provides the ideal control for testing antibody specificity for the endogenous protein in multiple applications. This capability then opens up the possibility of creating a standardized characterization process that can be applied systematically to characterize not only new antibodies but also the ~1 million antibodies that are already commercially available. With such a process in hand it should be possible to identify high quality antibodies for different applications from the existing set of commercially-available antibodies, seemingly for a large percentage of all human gene products.

To pilot the concept that excellent antibodies can be found among those that are commercially available if one carries out a systematic analysis, we studied the major amyotrophic lateral sclerosis (ALS, OMIM #105400) disease gene product, C9ORF72. ALS is a fatal neurodegenerative disease characterized by progressive paralysis leading to respiratory failure (*Kiernan et al., 2011*) and is on a genetic and pathophysiological continuum with frontotemporal dementia (FTD, OMIM #600274) (*Ng et al., 2015*). A search for genes involved in ALS/FTD led to the discovery of a hexanucleotide-repeat expansion mutation in the first intron of *C9ORF72*. In a North America cohort, this mutation underlies 11.7% and 23.5% of familial FTD and ALS cases, respectively, and in a large Finnish population, the *C9ORF72* mutation underlies 46.0% of familial ALS and 21.1% of sporadic ALS (*DeJesus-Hernandez et al., 2011*; *Renton et al., 2011*). Thus, the *C9ORF72* mutation is the most common genetic abnormality in both FTD and ALS. It is vital to understand the cell biological role of C9ORF72, but the literature regarding C9ORF72 subcellular and tissue-distribution is conflicting (*Burk and Pasterkamp, 2019*). We believe the lack of consensus on C9ORF72 expression, function and subcellular localization stems in part from the use of non-specific antibodies.

C9ORF72 provides an excellent protein on which to develop an antibody characterization process because although the protein is of relatively low abundance, there are many commercially-available antibodies. The process we outline can be applied to any protein target and here it led us to the recognition of problems with the C9ORF72 literature and to the discovery of unexpected properties of the protein.

## Results

### Development of an antibody validation strategy

The antibody validation strategy developed in this manuscript is presented in overview in *Figure 1*. The steps were built empirically as we proceeded with our analysis of antibodies for C9ORF72. The workflow is as follows: 1) use PaxDB (https://pax-db.org/) to identify a cell line that expresses the protein of interest at relatively high levels, is readily modifiable by CRISPR/Cas9, and is easy to grow

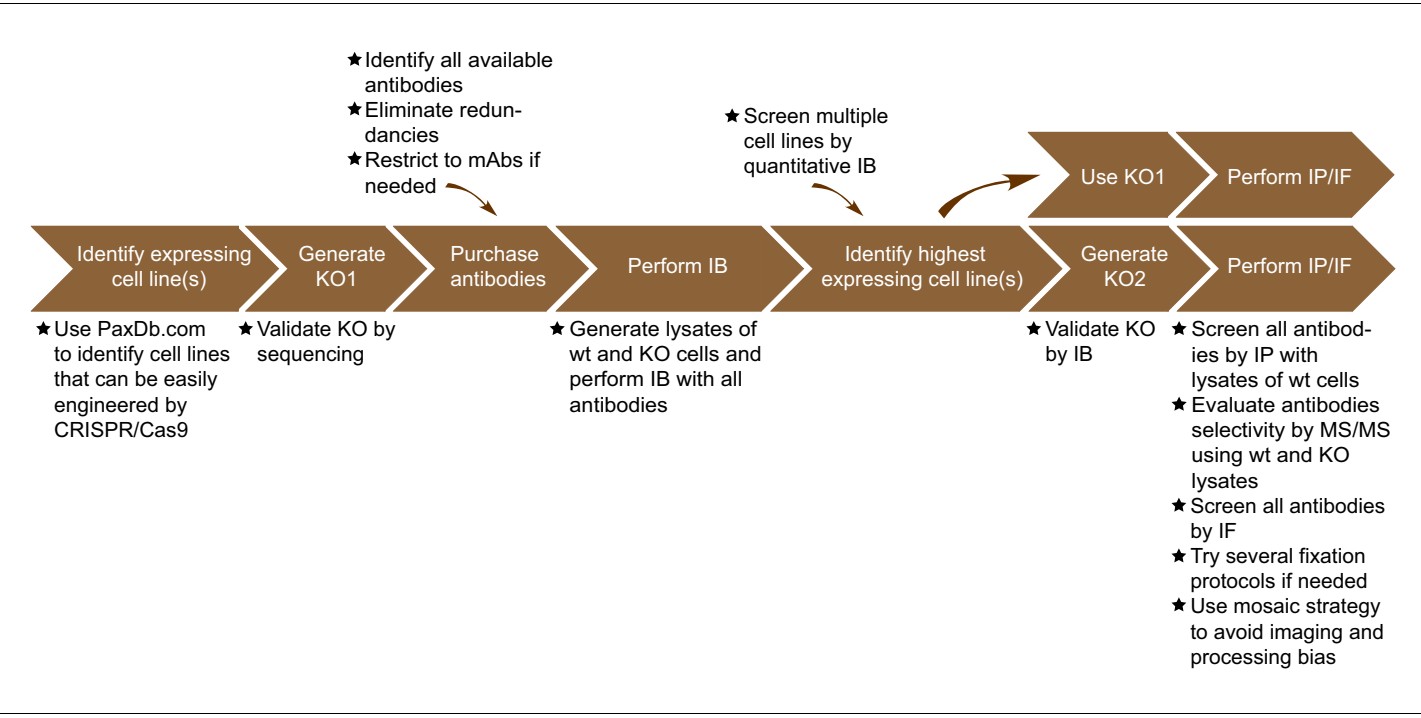

**Figure 1.** Summary of the antibody validation pipeline. IB = immunoblot; IP = immunoprecipitation; IF = immunofluorescence; KO = knockout.

and manipulate; 2) use CRISPR/Cas9 to generate a KO in this cell line; 3) identify all commercial antibodies against the protein of interest and screen them by immunoblot comparing the parental cell line to the KO controls; 4) use an antibody validated in step three for quantitative immunoblots on a panel of cell lines to identify a line that has the highest levels of expression of the target protein and has appropriate characteristics for the target of interest. This step is recommended as it provides more reliable information than could be provided by the proteomics database. However, the selected line may well be the original line that was chosen; 5) use the selected edited line to screen antibodies for specificity by immunoprecipitation and immunofluorescence.

## Selection and editing of a cell line and selection of antibodies

The single most common application of commercial antibodies for the majority of laboratories is immunoblot. We initially sought to screen all commercially available C9ORF72 antibodies by immunoblot comparing parental and KO cell lines. We first referenced PaxDb, a comprehensive proteomic database that provides protein abundance across multiple organisms and tissues/cell lines (*Wang et al., 2015*) (https://pax-db.org/). C9ORF72 is expressed at relatively low levels in most tissues and cell lines ranging from the 77th percentile for HeLa cells to the 35th percentile in RKO cells. We eventually chose to start with HEK-293 cells (expression levels at the 65th percentile) as they are the 5th most abundant C9ORF72 expressing cell line, and are easy to grow and to edit with CRISPR/Cas9. We then used CRISPR/Cas9 to generate C9ORF72 heterozygous and KO HEK-293 cells. Briefly, we introduced the sequence of our KO sgRNA into an expression plasmid bearing both the sgRNA scaffold backbone and Cas9. We then transfected the plasmid into HEK-293 cells and selected transfected cells with puromycin for 3 days. Cells were grown following clonal dilution and sequencing of the genomic DNA of multiple clones revealed double-strand breaks at the expected localization leading to one heterozygous and two complete KO lines.

We then purchased all known commercial antibodies for C9ORF72. For this we searched the literature by PubMed and examined the websites of companies with C9ORF72 antibodies identified using Google searches. We initially identified more than 100 antibodies for C9ORF72. However, we avoided purchasing the same antibody available from different companies. For example, the C9ORF72 antibody *Prestige Antibodies Powered by Atlas Antibodies* was purchased at Sigma (HPA023873), but is also sold by other companies including Novus Biological (NBP1-93504). This led

to 14 distinct commercial C9ORF72 antibodies, 10 rabbit polyclonal antibodies, three mouse monoclonal antibodies and one rabbit monoclonal antibody (*Supplementary file 1*). The antibodies were available from Abcam (5), Proteintech (3), GeneTex (3), Cell Signaling Technology (1), Sigma (1), and Santa Cruz (1). In addition we identified two sheep polyclonal antibodies from the Protein Phosphorylation and Ubiquitylation Unit of the Medical Research Council (MRC) in Dundee Scotland (*Supplementary file 1*). Interestingly, we eventually received refunds for many of the antibodies that did not perform as advertised, lowering overall costs.

## Analysis by immunoblot

We screened the 16 C9ORF72 antibodies by immunoblot using lysates of HEK-293 cells including parental cells, the isogenic heterozygous line and the two isogenic KO lines (*Figure 2*). For each series of blots we resolved 50 µg of protein (using 5–16% gradients gels) from lysates prepared in buffer containing 1% Triton X-100 in order to extract both cytosolic and membrane-associated proteins. The proteins, transferred to nitrocellulose membranes, were stained with Ponceau S to ensure even loading of lysates. After screening all 16 antibodies we found one mouse monoclonal antibody from GeneTex (GTX634482) that has optimal features (*Figure 2*). This antibody demonstrates a strong signal in parental HEK-293 cell lysates, the signal is reduced by ~50% in the heterozygous line, and no signal for the antibody is detected in the two KO lines (*Figure 2*). Moreover, no band other than C9ORF72 is visible, even upon long exposure (*Figure 2—figure supplement 1A*). Abcam rabbit monoclonal antibody (ab221137) and GeneTex mouse monoclonal antibody (GTX632041) also gave specific signals on immunoblot (*Figure 2*) but other non-specific bands were detected upon longer exposure (*Figure 2—figure supplement 1A*).

C9ORF72 is studied in mouse models of ALS and thus the identification of high-quality antibodies for murine C9ORF72 would be useful for the ALS community. We screened all 16 antibodies using brain extract from wild-type (WT) and C9ORF72 KO animals and found that both GTX634482 and ab221137 showed strong specific signals (*Figure 2—figure supplement 1A, B*). Other antibodies that recognize both human and murine C9ORF72 include ab12179 from Abcam, PT25757 and PT22637 from Proteintech, HPA023873 from Sigma, MRC-S478D and MRC-S479D from the MRC, and CST64196 from Cell Signaling Technology (*Figure 2*; *Figure 2—figure supplement 1B*). We were surprised to observe that some antibodies that did not detect C9ORF72 specifically in human lysates could identify C9ORF72 in mouse samples. These antibodies are GTX119776, ab227555 and sc-138763. All of these antibodies detect additional bands, and in several cases they reveal multiple cross-reactive species. Thus, GTX634482 from GeneTex and ab221137 from Abcam are robust antibodies for immunoblot.

## Analysis by immunoprecipitation

We next sought to test the functionality of the antibodies in immunoprecipitation applications. All 16 antibodies were pre-coupled to protein A- or protein G-Sepharose as appropriate and a detergent solubilized HEK-293 lysate from parental cells only was prepared and incubated with the antibody/bead mixtures. Controls included lysates incubated with beads alone and the bead/antibody conjugates incubated with buffer alone. After washing the beads, the presence of C9ORF72 in the immunoprecipitates was detected by immunoblot using rabbit antibody PT22637 for the immunoprecipitates with mouse antibodies, and the mouse antibody GTX634482 for immunoprecipitates with rabbit or sheep antibodies (*Figure 3*). Of the 16 antibodies, nine immunoprecipitate endogenous C9ORF72, with GeneTex monoclonal antibody GTX632041 demonstrating the most robust enrichment of the protein compared to starting material (SM) (*Figure 3A/B*). Seven antibodies showed no appreciable C9ORF72 immunoprecipitation including GTX634482 and ab221137 (*Figure 3A/B*), which were the most effective antibodies in immunoblot. We also determined the percentage of protein depleted from the supernatant after immunoprecipitation by performing immunoblots on the unbound fraction (*Figure 3—figure supplement 1*). For this we employed the LI-COR Odyssey Imaging System (LI-COR Biosciences), which utilizes fluorescent secondary antibodies to allow for quantitative immunoblots. Approximately 70% of endogenous C9ORF72 is captured from 1 mg of HEK-293 lysates using 1 µg of GTX632041, whereas the other 15 antibodies capture either no protein or at most 20% of total C9ORF72 (*Figure 3—figure supplement 1*). The

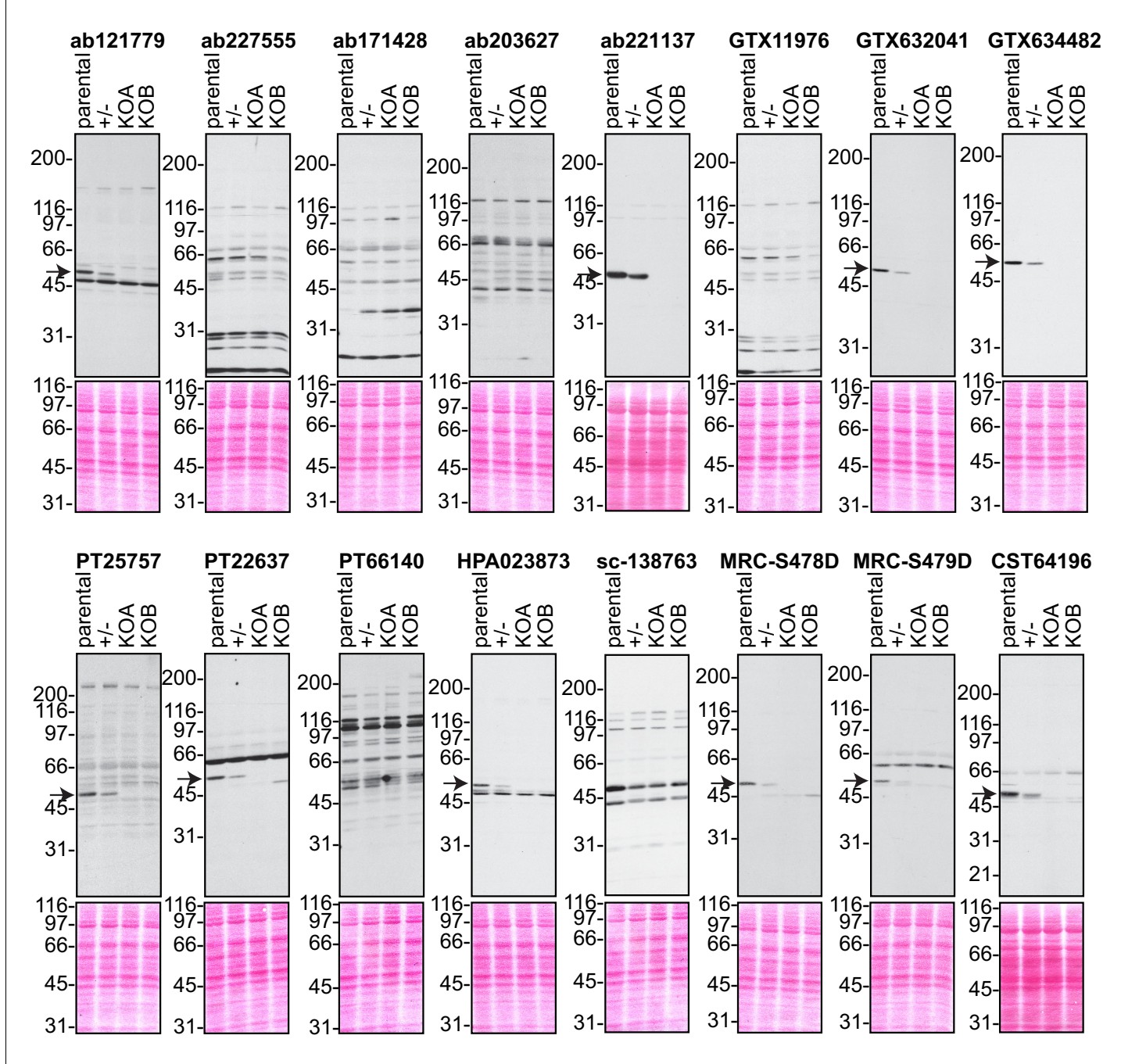

**Figure 2.** Analysis of 16 C9ORF72 antibodies by immunoblot. Cell lysates from HEK-293 parental, heterozygous (+/-) or two individual C9ORF72 KO clones (KOA, KOB) were prepared and processed for immunoblot with the indicated C9ORF72 antibodies. The arrows point to positive C9ORF72 signals. The Ponceau stained transfers associated with each blot are shown as a protein loading control.

The online version of this article includes the following figure supplement(s) for figure 2:

**Figure supplement 1.** Analysis of 16 C9ORF72 antibodies by immunoblot.

effectiveness of GTX634482 and ab221137 on immunoblot coupled to their inability to immunoprecipitate C9ORF72 highlights the need to screen all antibodies for all applications.

We next assessed the degree to which GTX632041 was selective for C9ORF72 in immunoprecipitation applications. We performed three independent immunoprecipitation studies with GTX632041 using parental and KO HEK-293 cell lysates (*Figure 3C*) followed by mass spectrometry analysis of

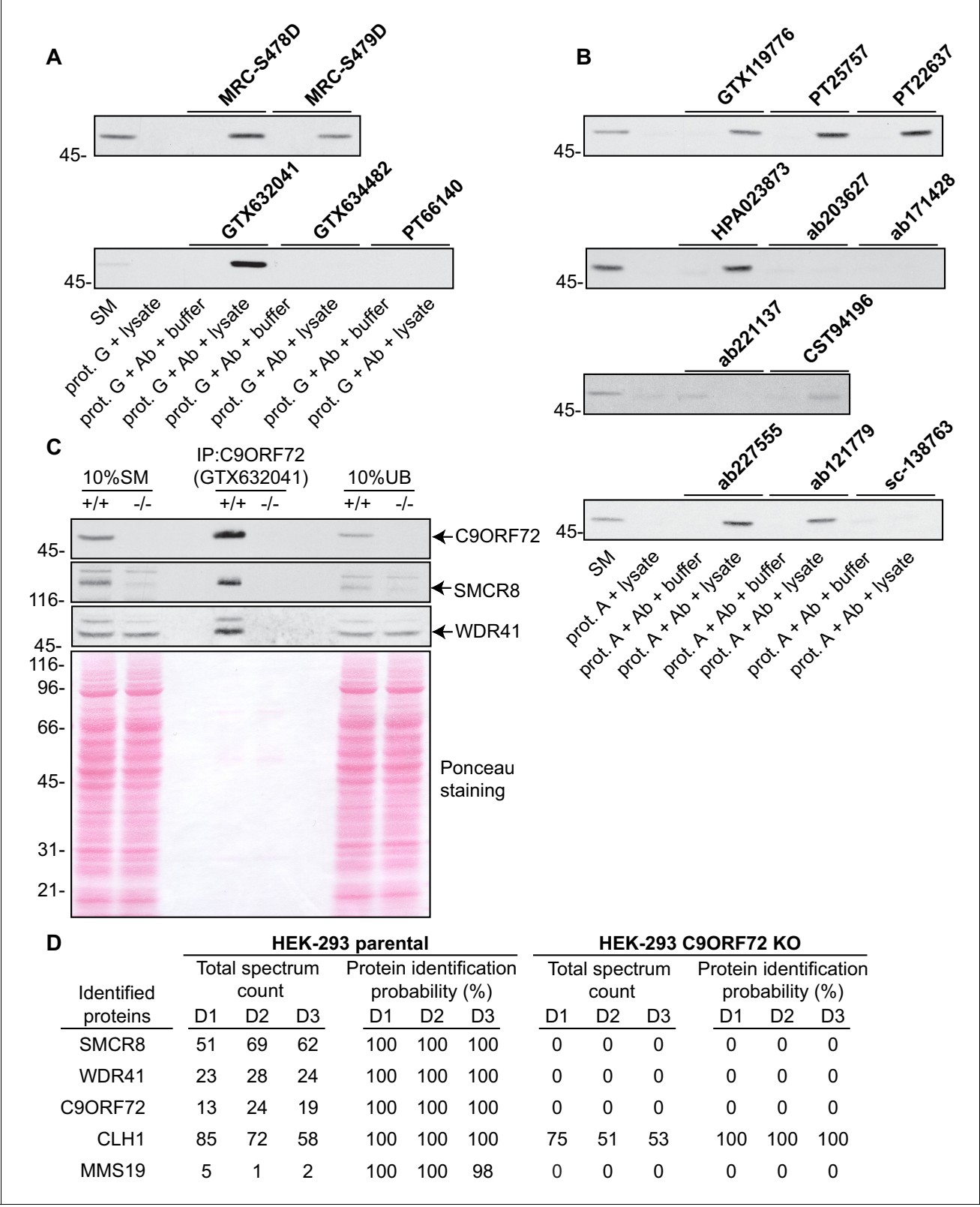

**Figure 3.** Analysis of 16 C9ORF72 antibodies by immunoprecipitation. (**A**) HEK-293 cell lysates were prepared and immunoprecipitation was performed using the indicated C9ORF72 antibodies pre-coupled to protein G-Sepharose (prot. G). Controls included the Protein G-Sepharose alone incubated with cell lysate or protein G-Sepharose pre-coupled with the antibodies but incubated with lysis buffer. Samples were washed and processed for immunoblot with C9ORF72 antibody PT22637. (**B**) HEK-293 cell lysates were prepared and immunoprecipitation was performed using the indicated

*Figure 3 continued on next page*

*Figure 3 continued*

C9ORF72 antibodies pre-coupled to protein A-Sepharose (prot. (**A**). Controls included the Protein S-Sepharose alone incubated with cell lysate or protein A-Sepharose pre-coupled with the antibodies but incubated with lysis buffer. Samples were washed and processed for immunoblot with C9ORF72 antibody GTX634482. (**C**) Lysates were prepared from HEK-293 cells, parental (+/+) and KO (-/-) and immunoprecipitation was performed using C9ORF72 antibody GTX632041 pre-coupled to protein G-Sepharose. Samples were washed and processed for immunoblot with C9ORF72 antibody PT22637. Blots were also performed for SMCR8 and WDR41. The Ponceau stained transfer of the blot is shown as a protein loading control. Aliquots (10%) of the cell lysates before (starting material, SM) and after incubation with the antibody-coupled beads (unbound, UB) were processed in parallel. (**D**) Table indicating the total spectrum count and protein identification probabilities for the indicated gene products proteins obtained by mass spectrometry analysis of the immunoprecipitated samples from three independent experiments (D1–D2–D3).

The online version of this article includes the following figure supplement(s) for figure 3:

**Figure supplement 1.** Comparison of the amount of C9ORF72 remaining in the unbound fraction after immunoprecipitation.

the immunoprecipitated samples (*Supplementary file 2*). Mass spectrometry of immunoprecipitates from the parental cells identified C9ORF72 (13, 24, 19 peptides) and two known C9ORF72-binding partners, SMCR8 (51, 69, 62 peptides) and WDR41 (23, 28, 24 peptides) (*Figure 3D*, *Supplementary file 2*) (*Sellier et al., 2016*; *Amick et al., 2016*; *Zhang et al., 2018*). The presence of both SMCR8 and WDR41 in the immunoprecipitates was confirmed by immunoblot (*Figure 3C*). The presence of C9ORF72 and its known-binding partners at near 1:1:1 level (when adjusting for the mass of the proteins) indicates that GTX632041 is selective for C9ORF72 and further supports the use of HEK-293 cells in our analysis. There were no peptides detected for any of these proteins in the immunoprecipitates from the C9ORF72 KO cells (*Figure 3D*, *Supplementary file 2*). Interestingly, we also identified MMS19, a 113 kDa protein component of the iron-sulfur protein assembly complex in the immunoprecipitates from parental cells (5, 1, 2 peptides) with no peptides in immuoprecipitates from the KO cells (*Figure 3D*), suggesting that MMS19 is a novel C9ORF72-binding partner. We also identified clathrin heavy chain (CLH1) with (85, 72 and 58 peptides) but similar numbers of peptides (75, 51, 53) were seen in the KO immunoprecipitations. This clearly indicates that the presence of clathrin heavy chain is not biologically significant but it is not possible to determine if it immunoprecipitates because it cross-reacts with the C9ORF72 antibody or for other non-specific reasons. In fact antibody cross-reactivity in immunoprecipitation applications is inherently difficult to determine. In addition to the proteins indicated in *Figure 3D* numerous other proteins were identified in the mass spectrometry studies but with limited numbers of peptides and always with similar peptide numbers from the parental and KO samples (*Supplementary file 2*).

## Analysis by immunofluorescence

We next tested the effectiveness of the antibodies for immunofluorescence applications. HEK-293 cells, parental and KO were fixed in paraformaldehyde (PFA), permeabilized with Triton X-100, and stained with C9ORF72 antibodies with appropriate fluorescent secondary antibodies. The antibodies gave various staining patterns with no reduction in fluorescent staining in KO cells (*Figure 4—figure supplement 1*). Similar results were seen when the cells were fixed in −20°C methanol (*Figure 4—figure supplement 2*).

Achieving successful detection of endogenous proteins by immunofluorescence can be challenging, especially for lower abundance proteins. Generating knockin lines with GFP tags on endogenous proteins could provide a mechanism to screen antibodies for such targets. In our study, we sought to obtain a better signal to noise ratio by identifying a cell line with higher levels of endogenous C9ORF72 expression. We thus used GTX634482 for quantitative immunoblots on multiple cell lines using the LI-COR system for detection. C9ORF72 is observed in a variety of human cell lines including U2OS (osteosarcoma), HeLa (cervical cancer), RKO (colon carcinoma), U87 and U251 (glioblastomas), motor neuron precursor cells (NPCs) and motor neurons (MNs) derived from human-induced pluripotent stem cells (*Figure 4—figure supplement 3A*). It is worth noting that with the LI-COR system (fluorescent secondary antibodies) the signals appear generally weaker than when using ECL development methods (compare the HEK-293 cell lysates from *Figure 4—figure supplement 3A* to those with GTX634482 in *Figure 2*). Even under these conditions there are additional bands detected in several of the cell lysates, indicating that even though GTX624482 is highly specific in HEK-293 cells and mouse brain, it recognizes non-specific bands in other cell lysates, an inherent weakness of any antibody validation procedure. Of the cell lines tested, U2OS has the highest levels

of C9ORF72 whereas RKO cells have lower expression than HeLa (*Figure 4—figure supplement 3A*). Interesting, the levels of C9ORF72 in motor neurons is similar to that in HEK-293. U2OS are easy to genome edit and are large cells suitable for immunofluorescence analysis. We thus used CRISPR/Cas9 to generate U2OS cells lacking C9ORF72, which was confirmed by immunoblot with GTX634482 (*Figure 4—figure supplement 3B*). To better distinguish between the parental and C9ORF72 KO lines, we transfected the lysosomal protein LAMP1 tagged to YFP in parental cells and LAMP1-RFP in C9ORF72 KO cells. We then re-plated the cells such that both parental and KO cells were found on the same coverslip as a mosaic (*Figure 4—figure supplement 3C*). This strategy reduces microscopy imaging and figure processing biases. We screened all 16 C9ORF72 antibodies using this strategy and we found GTX632041 as the sole antibody to recognize endogenous C9ORF72 specifically (*Figure 4A/B*). Indeed, GTX632041 immunofluorescence labeling reveals a cytosolic/punctate fluorescence signal that is dramatically reduced in KO cells to a level comparable to buffer control (*Figure 4A*). The other 15 antibodies showed non-specific staining patterns that are similar between control and KO cells (*Figure 4A/B*).

Based on this experience we recommend that once immunoblot screens identify a specific antibody for a target of interest, the next step should involve screening panels of cell lines with that antibody to find those with the highest expression levels (*Figure 1*). While PaxDb appears to be effective to identify appropriate lines to start the analysis it is not infallible. For example, while it indicated higher expression levels for C9ORF72 in U2OS versus HEK-293, it also indicated that RKO had higher levels than U2OS, which did not bear out in the immunoblot analysis (*Figure 4—figure supplement 3*). Thus, moving straight to KO in U2OS cells for the immunofluorescence analysis would have been a more effective strategy. Moreover, while the HEK-293 cells were effective for the immunoprecipitation studies here (*Figure 3*), we recommend also performing the immunoprecipitation studies following the cell distribution blots (*Figure 1*).

## Identification of C9ORF72 antibodies effective for immunohistochemistry

Given the importance of C9ORF72 in ALS and FTD it is vital to identify antibodies effective for staining of tissue sections. We thus used diaminobenzidine (DAB) labeling comparing brain sections from WT littermate mice to mice with the C9ORF72 ortholog (31100432O21Rik) deleted . We originally focused on GTX634482, which is the most specific antibody for immunoblot and recognizes the protein in mouse brain (*Figure 2—figure supplement 1A*). We tested the antibody on sections that had been treated at 110℃, pH 9.0 for epitope unmasking. A punctate and/or neuritic-like signal is observed in the neuropil of the glomerular layer of the olfactory bulb, the ventral pallidum of the basal ganglia, the CA4 (hilus), CA3, and CA2 region of the hippocampus, the substantia nigra, the inferior olive, and granular layer of the cerebellar cortex of WT mice (*Figure 5A*, arrows). The signal is nearly completely ablated in the KO mice although there is some non-specific signal seen in the dentate gyrus/CA4, inferior olive and cerebellar cortex sections (*Figure 5A*, arrows). The staining pattern observed is similar to the basal ganglia, hippocampal formation, and cerebellum staining observed in a previous study (*Frick et al., 2018*). GTX632041, which is the most effective antibody for immunoprecipitation (*Figure 3A*) and immunofluorescence (*Figure 4A*) analysis, also specifically recognizes C9ORF72 in mouse brain sections including those through the hippocampus (*Figure 5B*) but staining is not as strong or consistently specific as seen with GTX634482. Antibody ab221137, which worked well for immunoblot (*Figure 2*; *Figure 2—figure supplement 1A*) did not reveal any signal in the IHC analysis (*Figure 5C*).

Thus, we have identified three monoclonal antibodies that effectively and specifically recognize C9ORF72 in multiple applications. The antibody GTX634482 is recommended for immunoblot and immunohistochemical applications on antigen unmasked samples, and GTX632041 is recommended for immunoprecipitation and immunofluorescence. The antibody ab221137 is recommended for immunoblot.

## C9ORF72 localizes to lysosomes

The GeneTex antibodies GTX634482 and GTX632041 and the Abcam antibody ab221137 are valuable tools to assess C9ORF72 biology. We thus used these antibodies to resolve controversies regarding C9ORF72 tissue distribution and subcellular localization (*Burk and Pasterkamp, 2019*).

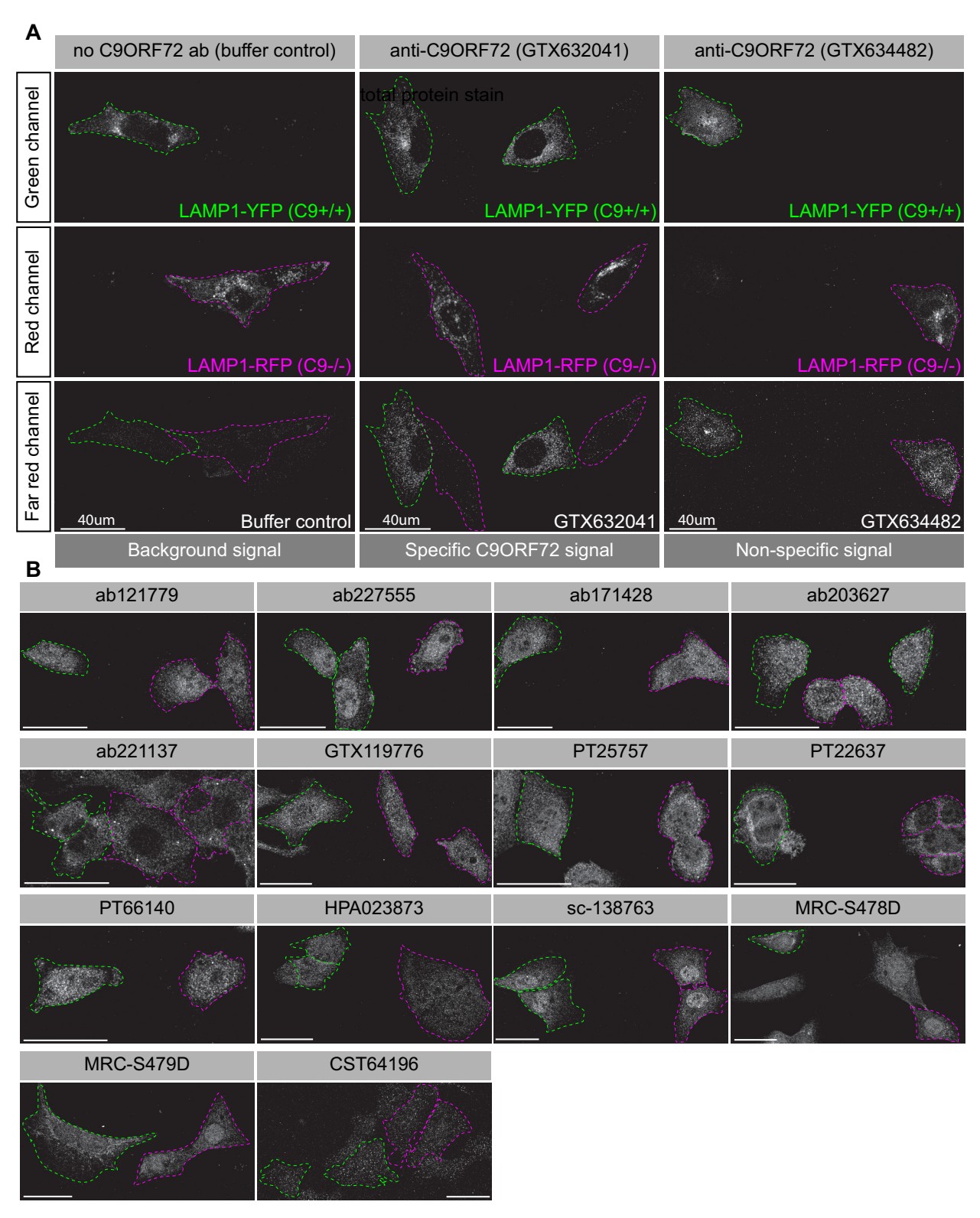

**Figure 4.** Analysis of 16 C9ORF72 antibodies by immunofluorescence. (**A**) Parental and KO cells were transfected with LAMP1-YFP or LAMP1-RFP, respectively. Parental and KO cells were combined and incubated with buffer only or stained with the indicated GeneTex C9ORF72 monoclonal antibodies. Grayscale images of the green, red and far-red channels are shown. Parental and KO cells are outlined with green and red dashed line, respectively. Representative images are shown. Bars = 40 μm. (**B**) Parental and KO cells were prepared as in (**A**) and stained with the indicated

*Figure 4 continued on next page*

*Figure 4 continued*

C9ORF72 antibodies. Grayscale images of the far-red channel are shown. Parental and KO cells are outlined with green and red dashed line, respectively. Representative images are shown. Bars = 40 μm.

The online version of this article includes the following figure supplement(s) for figure 4:

**Figure supplement 1.** Testing of C9ORF72 antibodies by immunofluorescence analysis in HEK-293 cells fixed with PFA HEK-293 parental and KO cells were fixed with PFA, permeabilized and incubated with the indicated C9ORF72 antibodies.

**Figure supplement 2.** Testing of C9ORF72 antibodies by immunofluorescence analysis in HEK-293 cells fixed with −20°C methanol HEK-293 parental and KO cells were fixed with −20°C methanol and incubated with the indicated C9ORF72 antibodies.

**Figure supplement 3.** immunofluorescence strategy in U2OS cells.

Endogenous C9ORF72 has been localized to the nucleus (*Renton et al., 2011*; *Atkinson et al., 2016*), early endosomes (*Farg et al., 2014*; *Shi et al., 2018*), recycling and late endosomes (*Farg et al., 2014*), the Golgi apparatus (*Aoki et al., 2017*), stress granules (*Chitiprolu et al., 2018*) the cytoplasm, neurites, growth cones (*Atkinson et al., 2016*), and lysosomes (*Shi et al., 2018*). Each of these studies performed immunofluorescence using commercial C9ORF72 antibodies that failed our validation criteria.

One study used CRISPR to add an HA tag to endogenous C9ORF72, localizing the fusion protein with an antibody against the tag. This study revealed a diffuse cytoplasmic localization for tagged C9ORF72 under basal cell culture conditions, with the observation that the protein became concentrated on lysosomes following starvation (*Amick et al., 2016*). We thus performed co-staining immunofluorescence studies with GTX632041 in cells expressing the lysosomal marker LAMP1 tagged with RFP. Under basal conditions, C9ORF72 demonstrates a consistent lysosomal localization (*Figure 6A*, top panels). As expected, starvation leads to larger LAMP1-positive lysosomes as a result of membrane fusion (*Yu et al., 2010*) but the localization of C9ORF72 is not altered (*Figure 6A*, bottom panels). In fact 82% of LAMP1-positive lysosomes show C9ORF72 puncta on their periphery in both fed and starved conditions (*Figure 6B*). Consistent with a lysosomal localization, C9ORF72 co-localizes in part with GFP-Rab7a and GFP-Rab9, two small GTPases involved in vesicle trafficking to lysosomes but has little co-localization with GFP-Rab5a or GFP-Rab11a, markers of early and recycling endosomes, respectively (*Figure 6—figure supplement 1A, B*).

To support the lysosomal localization of C9ORF72 observed by immunofluorescence, we analyzed lysosomes that were immuno-isolated using the lysosomal bait protein Tmem192-3xHA, a method previously described as a lysosomal immunoprecipitation (*Abu-Remaileh et al., 2017*). HEK-293 cells were transfected with Tmem192-3xHA (HA-Lyso cells) or with Tmem192-2xFlag (Control-Lyso cells). Following cell homogenization, an anti-HA immunoprecipitation was performed, leading to highly enriched lysosomes (*Abu-Remaileh et al., 2017*). This purification strategy resulted in the selective enrichment of LAMP1 by ~14 fold (*Figure 6C*). A mild enrichment of TOM20 was also observed as expected since lysosomes and mitochondria are tightly associated (*Wong et al., 2018*). C9ORF72 is enriched on lysosomes by ~3–4 fold in both basal and starved conditions (*Figure 6C*). In contrast, C9ORF72 is not enriched after immunoprecipitation of mitochondria (*Figure 6—figure supplement 2A*) using 3xHA-eGFP-OMP25 as bait (*Chen et al., 2016*).

While C9ORF72 definitively localizes to lysosomes a significant pool of the protein (~85%) is found outside of lysosomes (*Figure 6—figure supplement 2B*). When we performed a biochemical cytosol/membrane fractionation of RAW264.7 and U2OS cells followed by quantitative immunoblot, we found that ~75% of total C9ORF72 is in the soluble/cytosolic fraction in both cell lines, whereas 25% of total C9ORF72 is membrane-associated (*Figure 6—figure supplement 2C*). Thus, the fraction of C9ORF72 not found on lysosomes appears to correspond to a cytosolic pool. In conclusion, C9ORF72 distributes to a cytosolic pool and a membrane-bound pool on lysosomes.

## C9ORF72 localizes to late phagosomes and phagolysosomes in macrophages

C9ORF72 KO mice age without motor neuron disease (*O'Rourke et al., 2016*; *Jiang et al., 2016*; *Burberry et al., 2016*) but instead develop progressive splenomegaly and lymphadenopathy and ultimately die due to complications of autoimmunity (*O'Rourke et al., 2016*; *Burberry et al., 2016*; *Atanasio et al., 2016*). C9ORF72 is required for the normal function of murine myeloid cells, such as

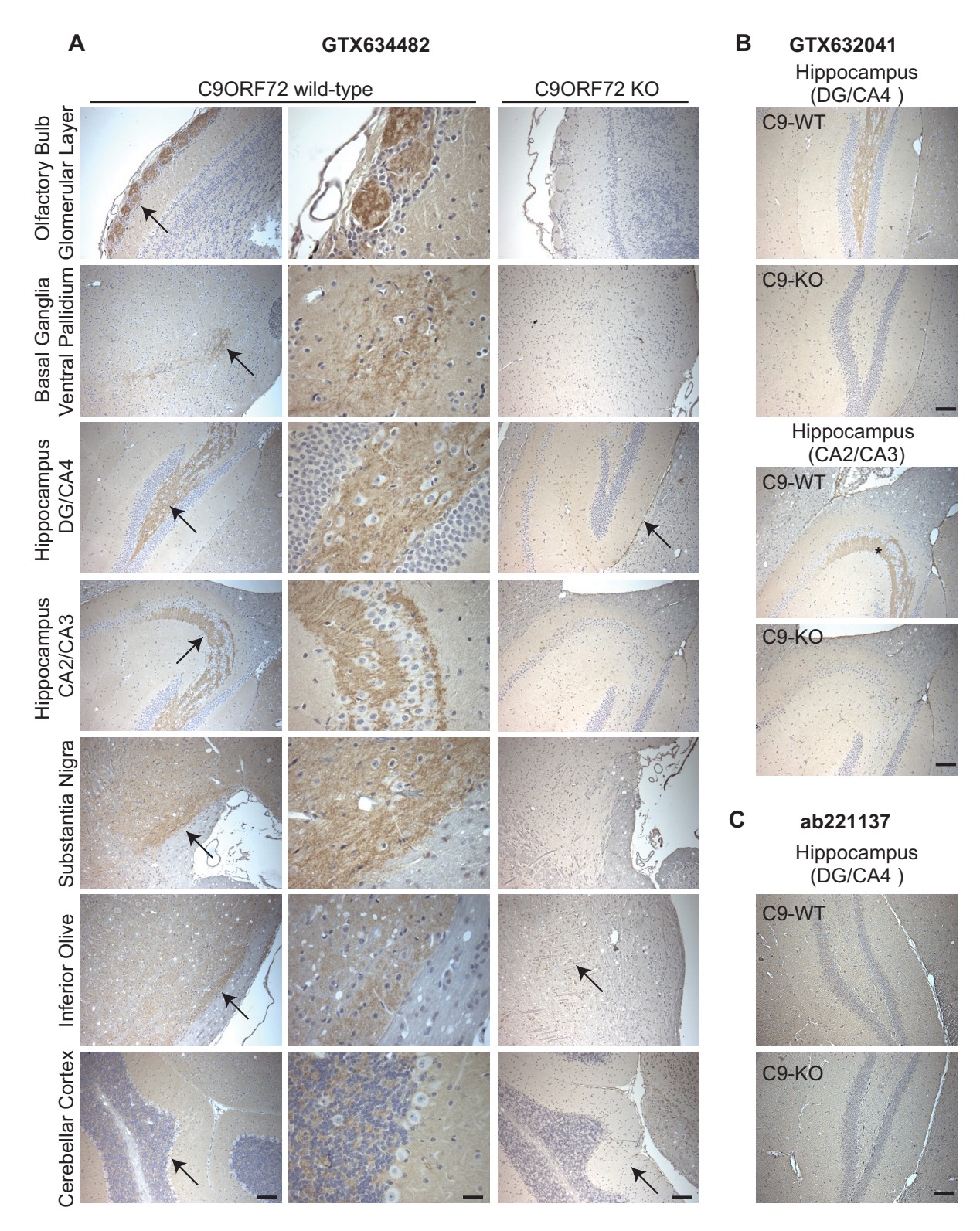

**Figure 5.** Analysis of C9ORF72 antibodies by immunohistochemistry. (**A**) DAB immunohistochemistry of C9ORF72 WT and C9ORF72 KO mouse brains in the sagittal plane using C9ORF72 antibody GTX634482. Micrographs are ordered from the rostral to caudal aspect of the mouse brain. For C9ORF72 WT, arrows indicate areas of increased DAB labeling intensity and correspond to the inset regions in the middle column of images. For C9ORF72 KO (right most column), arrows indicate areas of non-specific labeling. Scale bars = 100 µm or 25 µm for the inset images on the right for the C9ORF72 WT

*Figure 5 continued on next page*

*Figure 5 continued*

micrographs). (B) DAB immunohistochemistry of C9ORF72 WT and C9ORF72 KO mouse brains in the sagittal plane of the indicated brain regions using C9ORF72 antibody GTX632041. Scale bars = 100 μm. (C) DAB immunohistochemistry of C9ORF72 WT and C9ORF72 KO mouse brains in the sagittal plane of the indicated region using C9ORF72 antibody ab221137. Scale bar = 100 μm.

macrophages and microglia (resident macrophages in the central nervous system) (*O'Rourke et al., 2016*; *Atanasio et al., 2016*; *Jiang et al., 2016*) and C9ORF72 KO macrophages and microglia display lysosomal dysfunction (*O'Rourke et al., 2016*; *Zhang et al., 2018*). Interestingly, C9ORF72 is expressed at higher levels in bone marrow-derived macrophages (BMDM) from mice than in whole mouse brain or in cultures of rodent neurons (*Figure 7A*). Next, we examined human monocyte-derived macrophages (MDMs) under traditional M1 polarization conditions (IFN-γ and LPS treated) or in the presence of TGF-β, which reproduces the homeostatic state of the cells in vitro (*Healy et al., 2016*) (*Figure 7B*). We again observed strong expression of C9ORF72 in the macrophage cells.

Since the highest expression of C9ORF72 is in macrophages, we sought to investigate localization in these cells. In human MDMs, C9ORF72 decorates large organelles that appear to be phagosomes (*Figure 8A*). Macrophages are specialized phagocytic cells that engulf extracellular pathogens and cell debris into phagosomes. Early phagosomes are coated with F-actin (*Liebl and Griffiths, 2009*) before they mature into late phagosomes that lose their F-actin coat. Late phagosomes than fuse with lysosomes to form phagolysosomes. To confirm that C9ORF72 decorates phagosomes, we preceded with bead-capture experiments in RAW264.7 cells, a mouse macrophage-like cell line. First, RAW264.7 cells were differentiated from monocytes to macrophages by treating them with 1 μg/ml of LPS for 24 hr (*Figure 8B*). LPS treatment lead to an ~1.8 fold increase in C9ORF72 protein levels (*Figure 8C*). We then added 2.3 μm diameter beads and internalization was allowed for 30 min (pulse period) (*Figure 8D*). Cells were then washed and were chased (removal of the beads) for the indicated chase period. F-actin was used to label early-phagosomes, whereas LAMP1 was used to label phagolysosomes (*Figure 8D*). After a 30 min pulse (30-0), F-actin-coated early phagosomes are devoid of LAMP1, as expected, and devoid of C9ORF72. After a 30 min chase period (30-30), late phagosomes lost the majority of their actin coat, are devoid of LAMP1, but started to accumulate C9ORF72. A 90 min chase period (30-90) allowed enough time for late phagosomes to fuse with lysosomes. Phagolysosomes were devoid of F-actin and accumulated both LAMP1 and C9ORF72 (*Figure 8D*). Thus, C9ORF72 starts to accumulate on late-phagosomes once F-actin is being disassembled. C9ORF72 localizes before lysosome fusion on late-phagosomes and remains present on phagolysosomes.

In conclusion, we have developed an antibody validation process that is effective in identifying high-quality antibodies from pre-existing commercial pools and in revealing antibodies that should not be used or should be used with extreme caution. With high-quality antibodies new biology can emerge. The technologies employed are simple, easily applicable to any cell biological laboratory, and scalable. Most importantly, implementation of the procedures formalized in this study can serve as a template for granting agencies and journals as a mechanism to ensure enhanced reproducibility.

## Discussion

The scientific community, including academic scientists and commercial antibody producers recognize that there is a crisis in reproducibility caused by the use of non-validated antibodies. The problem could be alleviated with proper characterization of antibodies, but progress to this end has been impeded by technical, economic and sociological challenges. On the technological front, there are numerous types of antibodies (polyclonal, monoclonal, recombinant) used in many different applications - and each combination will require bespoke characterization. It is time-consuming and for some applications there is no quantitative readout. Moreover, these characterization processes are expensive and beyond the budget of any single antibody producer. Finally, there is no consensus among often conflicted scientists and companies, many of whom produce the antibodies, as to what the ideal antibody characterization pipeline should comprise. The advent of CRISPR genome-editing technology provides a powerful new approach to aid in antibody validation, and points to the

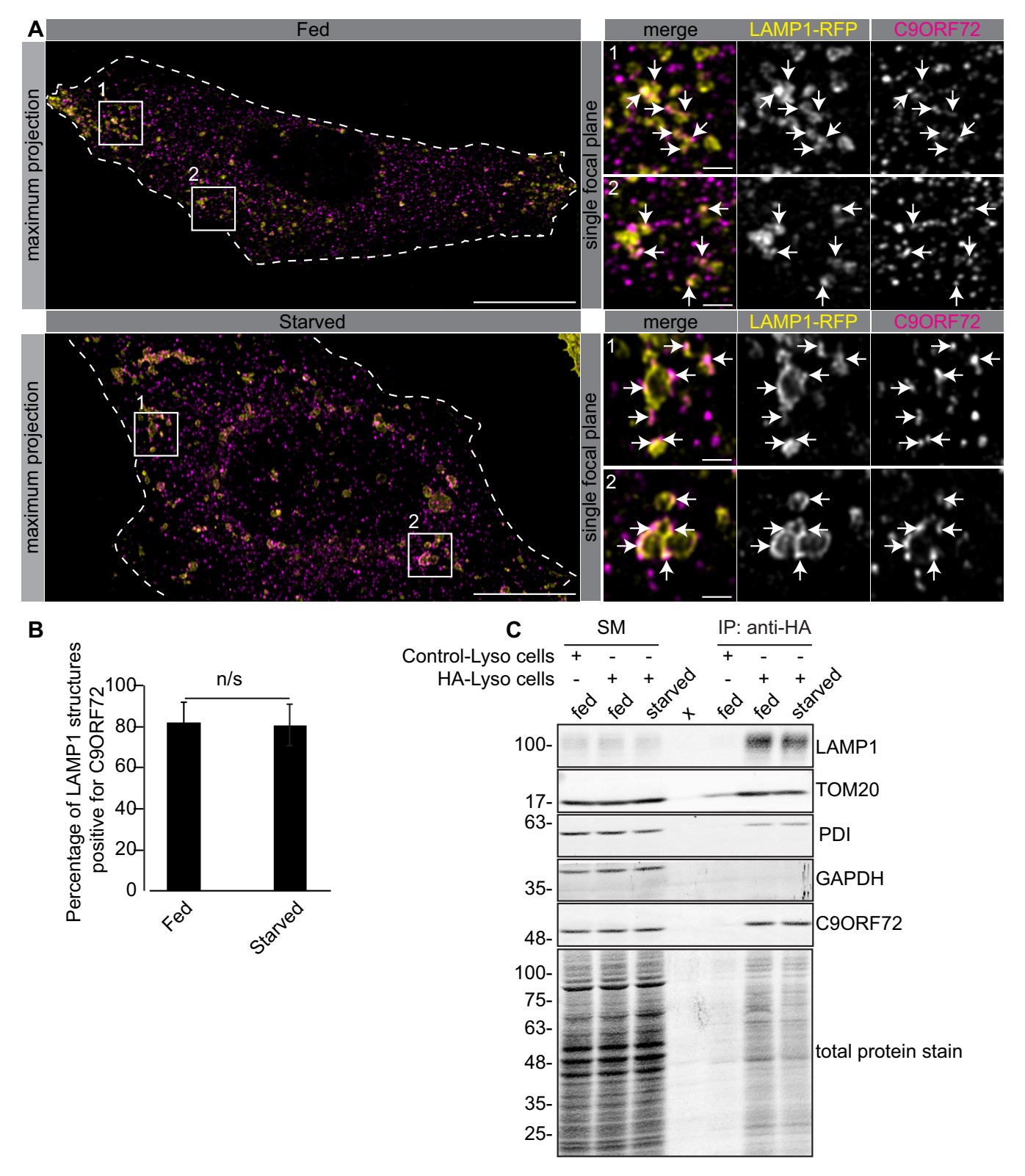

**Figure 6.** C9ORF72 localizes to lysosomes. (**A**) Immunofluorescence of endogenous C9ORF72 stained with GTX632041 in U2OS cells expressing LAMP1-RFP. The large images on the left show a maximum projection to highlight intracellular structures. Insets 1 and 2 are single focal plane, higher magnification images of the boxed regions. Merge and grayscale images of C9ORF72 and LAMP1-RFP are shown. C9ORF72 localization was observed under basal conditions (top panels) or under starvation for 2 hr (bottom panels). Cells are outlined with white dashed lines. Scale bars = 20 μm for large

*Figure 6 continued on next page*

*Figure 6 continued*

images and 2 μm for insets. (**B**) Quantification of LAMP1 structures positive for C9ORF72 in basal growing condition (fed) or 2 hr starved condition (starved) in U2OS cells. Between 228–262 lysosomes were counted from at least nine different cells from three independent experiments. A Student's t-test reveals that the percentage of LAMP1 structures positive for C9ORF72 is not significantly different between the fed and starved conditions (**C**) Lysates were prepared from HEK-293 cells expressing the Tmem192-3xHA (HA-Lyso cells) or the Tmem192-2xFlag (Control-Lyso cells). Lysosomes were immunoprecipitated using anti-HA magnetic beads. Quantitative immunoblotting for the indicated proteins was performed for starting material (SM), purified lysosomes and control immunoprecipitates (anti-HA IP). C9ORF72 was detected using GTX634482.

The online version of this article includes the following figure supplement(s) for figure 6:

**Figure supplement 1.** C9ORF72 co-staining with Rab.

**Figure supplement 2.** C9ORF72 subcellular fractionation.

possibility of creating a more standard antibody characterization approach that depends upon the use of KO controls.

Here we develop a simple, reproducible and relatively inexpensive approach for antibody characterization. While we applied the pipeline to all known commercial antibodies against C9ORF72, the process is certainly applicable to home-made antibodies and individual steps of the pipeline can be used in isolation depending on the needs of the user. Other complementary pipelines have been recently described (*Sikorski et al., 2018*). While our pipeline is for the most part straight forward, there are factors that need to be considered at each step. One important consideration is the choice of a target-expressing cell line to initiate the studies. We used PaxDb, a proteomic database providing protein abundance across several tissues/cell types (*Wang et al., 2015*) (https://pax-db.org/), which indicated that C9ORF72 is expressed in multiple cell lines (HEK-293, HeLa, U2OS) at equal or higher levels than brain, a result confirmed by quantitative immunoblot. PaxDb is useful but also limited in several respects. First, the reference mass spectrometry data is several years old but with continuing advances in proteomics and cell profiling approaches, it will becoming increasingly more accurate in general to predict appropriate cells lines for analysis. Moreover, PaxDb is limited to specific cell lines. In this context, it is important to avoid preconceived notions regarding protein distribution. For example, it is often assumed that neurodegenerative disease genes are predominantly expressed within the neuronal population that is most sensitive to degeneration, yet it is emerging that this is often not the case. Indeed very few are expressed specifically in the nervous system and most are found in multiple cell and tissue types (see the Genotype-Tissue Expression Project; https://gtexportal.org/home/). One example that parallels our findings for C9ORF72 is the major Parkinson disease gene LRRK2 that is broadly expressed in non-neuronal cells, and even within the nervous system, LRRK2 expression levels are low in the most affected area, the substantia nigra (*Rui et al., 2018*). In fact LRRK2 is most highly expressed in immune cells (*Dzamko, 2017*). Our study reveals that the highest levels of C9ORF72 are in macrophages and yet we could not determine this with PaxDb as the database has no proteomic data on this cell type. Therefore, an immunoblot-validated antibody should be used early in the pipeline to screen multiple cell lines.

Another factor to consider are the redundancies seen in commercial antibodies as companies cross-license. For C9ORF72 we identified over 100 commercial antibodies from multiple companies, but when eliminating redundancies, we recognized 16 unique antibodies. While this analysis was tedious, the process could be improved by searches involving Research Resource Identifiers (RRID) that provide unique and permanent identifiers to every antibody (https://scicrunch.org/resources). The number of antibodies available for each target is variable with, for example, eight antibodies for the newly identified ALS disease gene NEK1 and over 700 for SOD1, an abundant and well-studied ALS gene. If the number of antibodies is unmanageable, it would be advisable to limit validation studies to renewable antibodies such as monoclonals (*Andrews et al., 2019*). Also, whereas the cost of antibody purchase can appear to be a barrier, many companies have refund policies that can decrease costs if the antibodies to not meet reasonable functional criteria.

Perhaps the most important considerations are the conditions for the analysis by immunoblot, immunoprecipitation, and immunofluorescence. For immunoblot these involve but are not limited to the method of cell lysis, the amount of material analyzed, the type of gel, the transfer method, and the immunoblot conditions. For immunoprecipitation the cell lysis conditions are particularly important as it is vital to generate a fully soluble lysate that extracts the target protein yet maintains native structure. Perhaps the most challenging procedure is immunofluorescence. In our study we were

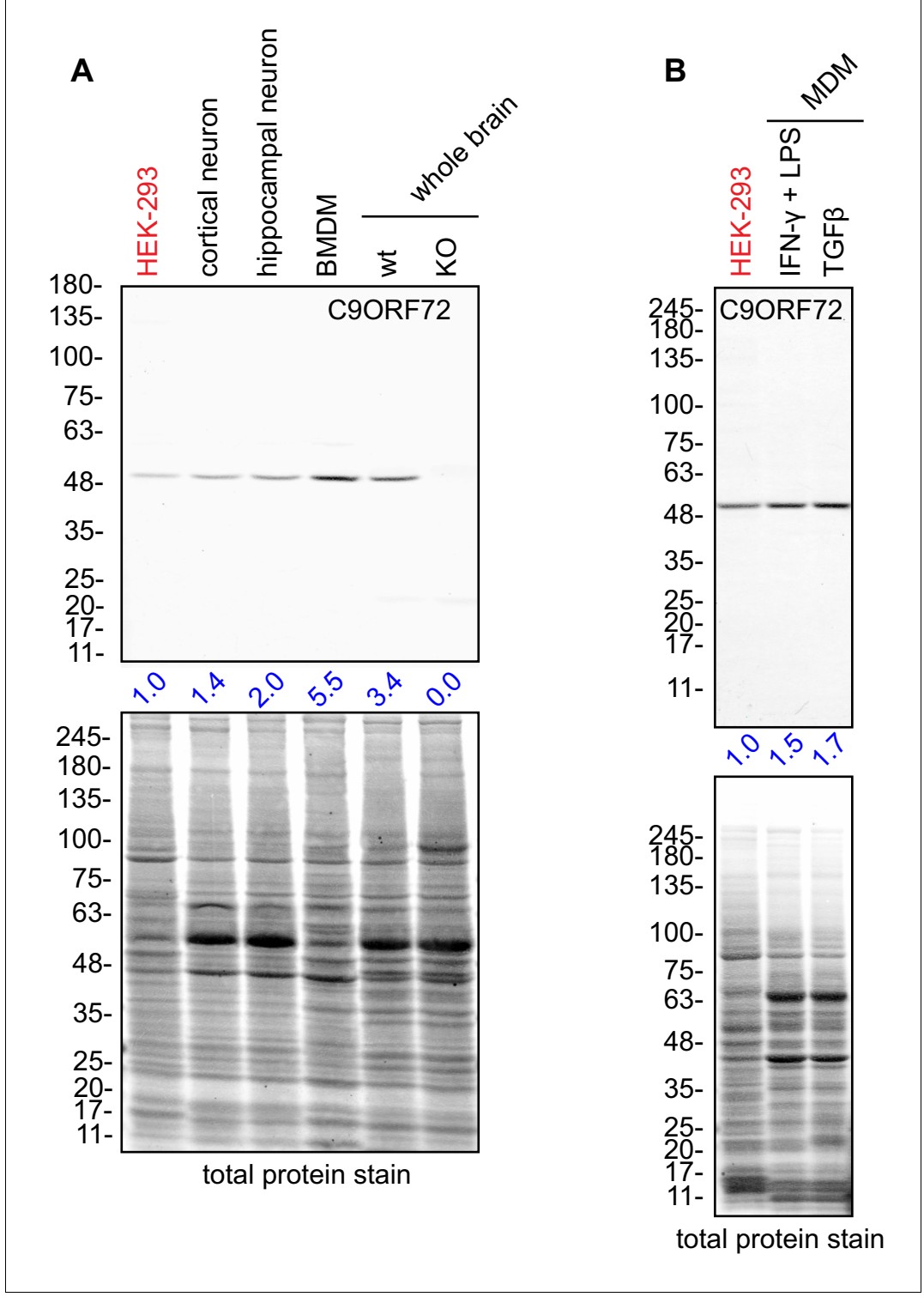

**Figure 7.** Macrophages express C9ORF72 at high levels. (**A**) Lysates were prepared from rat cortical neurons, rat hippocampal neurons, mouse bone marrow-derived macrophages (BMDMs) and mouse whole brain, and processed for quantitative immunoblot using antibody GTX634482 with the LI-COR Odyssey Imaging System that utilizes fluorescent secondary antibodies to allow for quantitative blots. The total protein stained transfers are shown as a loading control. The C9ORF72 protein signal as a ratio to total protein was determined, normalized to parental HEK-293 cells (red), and presented as fold change (blue numbers). C9ORF72 KO mouse brain (KO) was included as a specificity control. (**B**) Lysates were prepared from human monocyte-derived macrophages (MDMs)

*Figure 7 continued on next page*

*Figure 7 continued*

treated with either IFN-γ and LPS or with TGF-β. Immunoblot and C9ORF72 quantification were performed as in (**A**).

unable to observe a specific signal in HEK-293 cells despite multiple attempts in which we varied fixation and permeabilization conditions. However, when we analyzed U2OS and macrophages, which have higher expression levels, we obtained a specific signal with little effort using 4% PFA followed by Triton X-100 permeabilisation. In a multi-variant screen of fixation conditions for immunofluorescence, it was concluded that 4% PFA followed by Triton X-100 permeabilisation will suffice for the vast majority of targets (*Stadler et al., 2010*). Therefore, we stress that one needs to perform the immunofluorescence studies in parental versus KO cells after determining an appropriate, highly expressing cell line by immunoblot to ensure a high signal-to-noise ratio (*Figure 1*). Given that the human proteome contains a vast range of proteins with highly variant chemical compositions, it is difficult to envision set protocols for any of these procedures. It is a challenge to the pipeline to determine empirically the appropriate parameters and this is best left to Investigators with knowledge of selected targets of interest.

It is relatively straight forward to evaluate antibody performance by immunoprecipitation followed by immunoblot in terms of antibody effectiveness in precipitating the target. Moreover, by analyzing the unbound material one can achieve a good sense of efficiency of immunoprecipitation. For example, GTX632041 significantly enriched C9ORF72 in the immunoprecipitate by capturing approximately 70% of all available C9ORF72 in our experimental setup. However, it is much more difficult to determine antibody specificity for the intended target. Analysis by mass spectrometry revealed hundreds of proteins in the GTX632041 immunoprecipitates from parental HEK-293 cells. While the vast majority of these proteins were also detected in immunoprecipitates from the C9ORF72 KO cells, indicating that they are not relevant C9ORF72-interacting proteins, it is almost impossible to tell whether they cross-react with GTX632041 or are present through another forms of non-specific interactions. However, we can stipulate that a protein identified in the parental immunoprecipitates that is absent from the KO immunoprecipitates is a *bona fide* C9ORF72 binding partner. C9ORF72, SMCR8 and WD41 are found with robust peptide counts in the parental immunoprecipitates and absent in the KO immunoprecipitates. MMS19 was also identified only in the parental immunoprecipitates but with low peptide counts, suggesting a novel, transient interaction with the C9ORF72 protein complex.

Finally, it is important to mention that all antibodies need to be tested under all conditions since the performance and specificity of an antibody depends on the nature of the binding to its protein target, including if the recognition involves linear or conformational epitopes, which must be determined empirically (*Forsström et al., 2015*). A clear example in this study is GTX634482 that has optimal properties for immunoblot and immunohistochemistry but is not specific in immunofluorescence and is totally negative in immunoprecipitation.

The two GeneTex mouse monoclonal antibodies effective for immunoblot, immunoprecipitation, immunohistochemistry and immunofluorescence and the Abcam rabbit monoclonal ab221137 effective for immunoblot have not yet been used in a published paper, whereas the GeneTex rabbit polyclonal antibody (GTX119776), which does not detect the protein by blot and only weakly immunoprecipitates has been used in five published papers (*Supplementary file 1*). A troubling finding of our study is that the rabbit polyclonal antibody SC-138763, which clearly does not recognize C9ORF72 in any human sample in any application, has been used in 15 published manuscripts to ascribe specific properties to the protein in normal and disease states (*Supplementary file 1*). A Google Scholar-based bibliometric approach revealed that those 15 papers have been cited greater than 3000 times and that first layer citations to those citing papers is greater than 66000 citations. While clearly the majority of first and second layer citations do not focus on discoveries related to the use of the antibody in the original 15 papers, this simple analysis reveals the cumulative and sustained impact of inadequate characterization of antibodies.

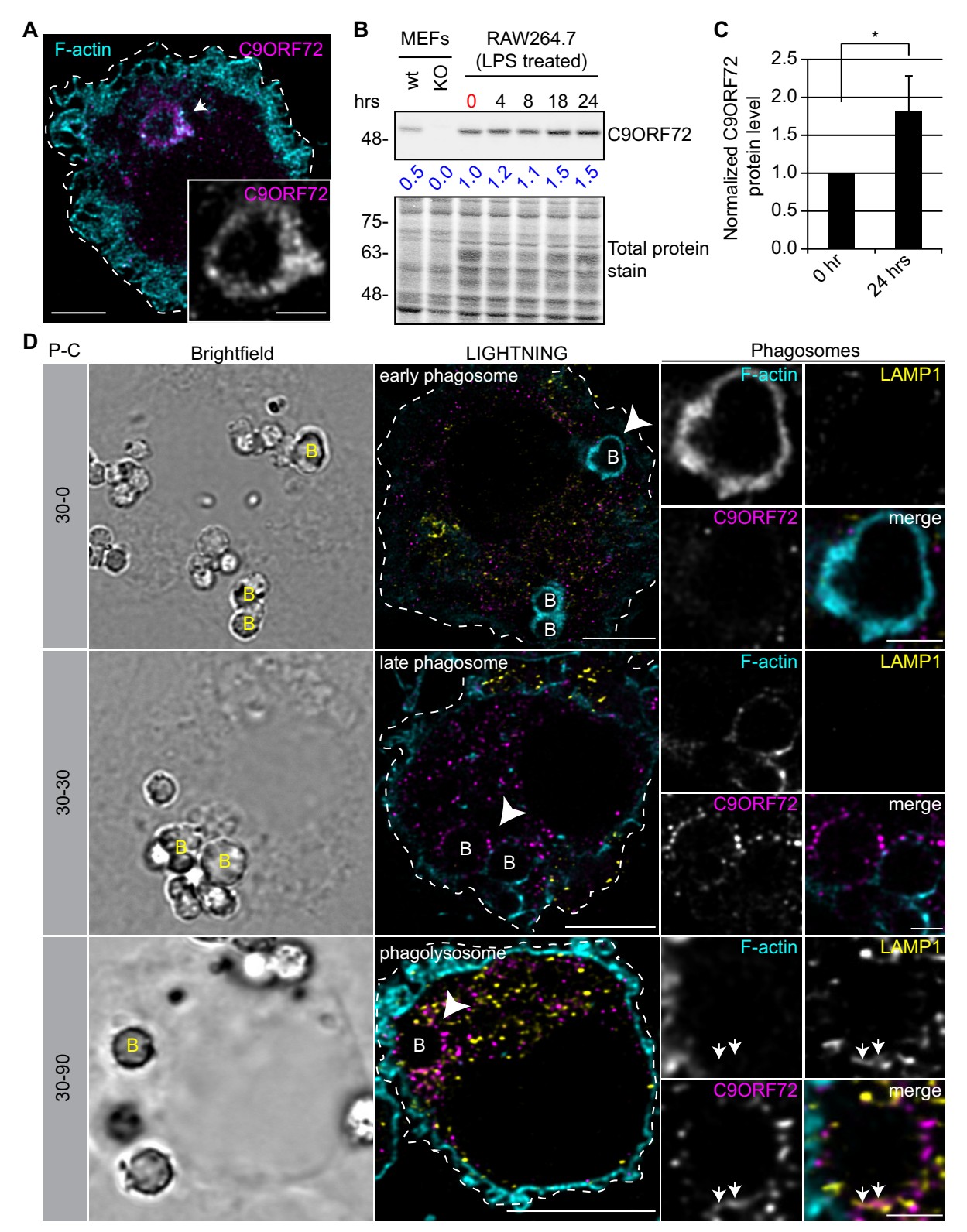

**Figure 8.** C9ORF72 localizes in part to late phagosomes and phagolysosomes.  (**A**) Immunofluorescence of C9ORF72 (GTX632041, magenta) and F-actin staining (cyan) in human MDMs. The inset is a higher magnification of the region indicated by the arrow and shows a grayscale image of C9ORF72. Cell is outlined with a white dashed line. Scale bar = 5 µm (full size) and 2 µm (inset). (**B**) Quantitative immunoblot of RAW264.7 cell lysates treated with 1 µg/ml LPS for the indicated time points. The total protein stained transfer is shown as loading control. C9ORF72 is detected using

*Figure 8 continued on next page*

**Figure 8 continued**

GTX634482. The C9ORF72 protein signal as a ratio to total protein was determined, normalized to RAW264.7 lysates at time 0 (red), and presented as fold change (blue numbers). C9ORF72 KO MEFs are included as a specificity control. (**C**) Quantification of C9ORF72 intensity from immunoblots performed as in (**B**) at 0 hr and 24 hr time points, presented as normalized values. Four individual experiments were quantified. *=p value<0.05 (**D**) Pulse-chase (**P–C**) experiments in LPS-treated RAW264.7 cells using phagocytosed beads. Beads were incubated for a 30 min pulse followed by a 0 min (30-0), 30 min (30-30) or 90 min (30-90) chase. Brightfield imaging reveals the phagocytosed beads. Staining of C9ORF72 (GTX632041; magenta), LAMP1 (yellow) and F-actin (cyan) were performed and imaged using confocal microscopy together with LIGHTNING image processing as described previously. Single focal plane images are shown. Insets are higher magnification views of the internalized beads indicated with arrowheads in the large images and show the merged channel together with grayscale images of F-actin, LAMP1 and C9ORF72. White arrows point toward C9ORF72 and LAMP1 co-localization. All images are the result of LIGHTNING applied to the confocal image. Cells are outlined with white dashed lines. Scale bar = 8 µm (full size) and 2 µm (inset).

## Materials and methods

### Antibodies

All C9ORF72 antibodies are listed in *Supplementary file 1*. SMCR8 antibody is from Abcam (ab202283). WDR41 antibody is from Abgent (AP10866B-EV). TOM20 antibody is from Santa Cruz Biotechnology, Inc (sc-11414). PDI antibody is from Abcam (ab2792). GAPDH antibody is from Ori-Gene (TA802519). LAMP1 antibody used for immunoblot is from Cell Signaling Technology (#9091). LAMP1 antibody used for IF in RAW264.7 cells is from Thermo (PA1-654A). Peroxidase-conjugated goat anti-mouse and anti-rabbit are from Jackson ImmunoResearch Laboratories. Odyssey IRDye 800CW goat anti-mouse-HRP, the REVERT total protein stain solution and the Odyssey Blocking buffer (TBS) are from LI-COR Biosciences. Alexa Fluor 647-conjugated mouse and rabbit secondary antibodies are from Invitrogen.

### Mouse breeding

Mice were bred and cared for in accordance with the guidelines of the Canadian Council on Animal Care following protocols approved by the University of Toronto Animal Care Committee. C9ORF72 KO mice were a generous gift of Dr. Don Cleveland (UCSD) and Dr. Clothilde Lagier-Tourenne (UMass). In-house breeding and genotyping were performed as previously described (*Jiang et al., 2016*). Briefly, heterozygous C9ORF72 mice (C57BL/6 background) were crossed to produce homozygous C9ORF72 KO mice, heterozygous C9ORF72 mice, and WT littermates.

### Cell lines

HEK-293 cell line is from ATCC (CRL-1573). U2OS cell line is from ATCC (HTB-96). Once received from ATCC, cells were immediately amplified and froze down in multiple aliquots. Generation of KO in HEK-293 and U2OS lines was done using low passage cells. The RAW264.7 cell line is from American Type Culture Collection, Rockville, MD (*Duclos et al., 2000*). Cell lines are monthly tested for mycoplasma contamination using the mycoplasma detection kit (biotool cat# B39038). Contaminated cell lines are immediately discarded.

### Cell culture

HEK-293 were cultured in DMEM high-glucose (GE Healthcare cat# SH30081.01) containing 10% bovine calf serum (GE Healthcare cat# SH30072.03), 2 mM L-glutamate (Wisent cat# 609065, 100 IU penicillin and 100 µg/ml streptomycin (Wisent cat# 450201). U2OS were cultured in DMEM high-glucose containing 10% tetracyclin-free fetal bovine serum (FBS) (Wisent cat# 081150) 2 mM L-glutamate, 100 IU penicillin and 100 g/ml streptomycin. Tetracyclin-free FBS was used to limit Cas9 expression. U2OS were starved for 2 hr in Earle's balanced salts medium (Sigma cat# E2888).

### CRISPR/Cas9 genome editing

For the KO HEK-293 cells, a C9ORF72 KO gRNA was annealed and ligated into Bbs1-digested pSpCas9(BB)−2A-Puro (PX459) V2.0 vector. PX459 V2.0 was a gift from Feng Zhang (Addgene plasmid # 62988). The gRNA was designed using the 'optimized CRISPR design' tool (www.crisp.mit.edu). Oligonucleotides with Bsb1 cleavage overhangs were annealed and cloned into the Cas9/puromycin expressing vector (PX459 from Addgene #48139). The following gRNA sequence was used to

create the KO line: CAACAGCTGGAGATGGCGGT. Genome editing plasmids expressing gRNA were transfected in HEK-293 cells using jetPRIME Transfection Reagent (Polyplus) according to the manufacturer's protocol. The next day, transfected cells were selected using 2 µg/ml puromycin for 24 hr. At 96 hr post-selection, cells were isolated by clonal dilution. Following the selection expansion of colonies, KOs were confirmed by sequencing of PCR-amplified genomic DNA (data not shown) and immunoblot. Genomic DNA was extracted using QuickExtract DNA extraction solution (Epicentre Biotechnologies).

For the KO U2OS cells, we first generated a line with stable-inducible Cas9. The line was generated using the pAAVS1-PDi-CRISPRn knockin vector and the TALEN pair arms were gifts from Bruce Conklin (Addgene plasmid # 73500) (*Mandegar et al., 2016*). U2OS cells were resuspended in complete media and 1 × 10$^5$ cells were mixed with pAAVS1-PDi-CRISPRn knockin vector (0.25 ug) and each AAVS1 TALEN pair (0.1 ug) and nucleofected using the Neon Transfection System (program 17; Life Technologies). Polyclonal population was selected with 2 µg/ml of puromycin (Bioshop cat# PUR555). Seventy-two hours post-treatment, single cells were plated in a 96-well plate. U2OS-PDi-CRISPRn positive clones were detected by PCR and the presence of the PDi-CRISPRn knockin platform into the safe-harbor AAVS1 locus was confirmed by Sanger sequencing. To KO C9ORF72, synthetic single gRNAs (sgRNAs) were designed using the CRISPR Design Tool from Synthego (https://www.synthego.com/products/bioinformatics/crispr-design-tool). The sgRNAs (from Synthego) were transfected in the stable-inducible Cas9 U2OS cell line using JetPRIME transfection reagent. At 1 hr post transfection, Cas9 expression was induced by treating cells with 2 µg/ml of doxycycline (Bioshop cat# DOX44). The next day, media was changed to remove doxycycline and stop the expression of Cas9. C9ORF72 KO cells were isolated by clonal dilution and KOs were confirmed by sequencing of PCR-amplified genomic DNA and immunoblot. Genomic DNA was extracted using QuickExtract DNA extraction solution (Epicentre Biotechnologies). C9ORF72 targeting sgRNA1: GCAACAGCUGGAGAUGGCGG, C9ORF72 targeting sgRNA2: GUCUUGGCAACAGCUGGAGA, AAVS1 locus targeting sgRNA1.2: GGGGCCACUAGGGACAGGAU and AAVS1 locus targeting sgRNA 1.3: GUCCCCUCCACCCCACAGUG. Cell lines transfected with AAVS1 sgRNA1.2 and AAVS1 sgRNA 1.3 are called parental A and parental B, respectively.

## Immunoblot

Cultured cells were collected in HEPES lysis buffer (20 mM HEPES, 100 mM sodium chloride, 1 mM EDTA, 5% glycerol, 1% Triton X-100, pH 7.4) supplemented with protease inhibitors. Following 30 min on ice, lysates were spun at 238,700xg for 15 min at 4°C and equal protein aliquots of the supernatants were analyzed by SDS-PAGE and immunoblot. Brains from 3 month old mice were homogenized in HEPES lysis buffer (without detergent) using a glass/Teflon homogenizer with 10 strokes at 2000 rpm and Triton X-100 was added to 1% final concentration. Following 30 min on ice, mouse tissues lysates were spun at 238,700xg for 15 min at 4°C. Equal protein aliquots of the supernatants were analyzed by SDS-PAGE and immunoblot. Immune cell populations were prepared from spleen, thymus and bone marrow by disrupting the tissue in media using a sterile syringe plunger as a pestle and passing the slurry through a 40 µm cell strainer. Red blood cells from were removed using a red blood cell lysis buffer (155 mM NH$_4$Cl, 12 mM NaHCO$_3$, 0.1 mM EDTA). Equal protein aliquots of the supernatants were analyzed by SDS-PAGE and immunoblot.

Immunoblots were performed with large 5–16% gradient polyacrylamide gels and nitrocellulose membranes. Proteins on the blots were visualized by Ponceau staining. Blots were blocked with 5% milk, and antibodies were incubated O/N at 4°C with 5% bovine serum albumin in TBS with 0.1% Tween 20 (TBST). The peroxidase conjugated secondary antibody was incubated in a 1:10000 dilution in TBST with 5% milk for 1 hr at room temperature followed by washes. For quantitative immunoblots, nitrocellulose transfers were incubated with REVERT total protein stain to quantify the amount of protein per lane, than blocked in Odyssey Blocking buffer (TBS) and C9ORF72 antibody GTX634482 was incubated O/N at 4°C in TBS, 5% BSA and 0.2% Tween-20. The secondary antibody (Odyssey IRDye 800CW) was incubated in a 1:20000 dilution in TBST with 5% BSA for 1 hr at room temperature followed by washes. Detection of immuno-reactive bands was performed by image scan using a LI-COR Odyssey Imaging System (LI-COR Biosciences) and data analysis was done using LI-COR Image Studio Lite Version 5.2.

## Immunoprecipitation

HEK-293 cells were collected in HEPES lysis buffer supplemented with protease inhibitors. Following 30 min on ice, lysates were spun at 238,700xg for 15 min at 4°C. One ml aliquots at 1 mg/ml of lysate were incubated for ~18 hr at 4°C with either 1 µg of a C9ORF72 antibody coupled to protein A or G Sepharose. Beads were subsequently washed four times with 1 ml of HEPES lysis buffer, and processed for SDS-PAGE and immunoblot. For immunoprecipitation with GTX632041 prior to mass spectrometry, HEK-293 cells (parental and C9ORF72 KO) were collected in HEPES lysis buffer supplemented with protease inhibitors. Following 30 min on ice, lysates were spun at 238,700xg for 15 min at 4°C. One ml aliquots at 1 mg/ml were incubated with empty protein G Sepharose beads for 30 min to reduce the levels of proteins bound non-specifically. These pre-cleared supernatants were incubated for 4 hr at 4°C with GTX632041 antibody coupled to protein G Sepharose. Following centrifugation, the unbound fractions were collected and the beads were washed 3 times with 1 ml of HEPES lysis buffer. The beads were then suspended in 1X SDS gel sample buffer. Fractions of the sample were processed for immunoblot. Parallel fractions were run into a single stacking gel band on SDS-PAGE gels to remove detergents and salts. The gel band was reduced with DTT, alkylated with iodoacetic acid and digested with trypsin. Extracted peptides were re-solubilized in 0.1% aqueous formic acid and loaded onto a Thermo Acclaim Pepmap (Thermo, 75 uM ID X 2 cm C18 3 uM beads) precolumn and then onto an Acclaim Pepmap Easyspray (Thermo, 75 uM X 15 cm with 2 uM C18 beads) analytical column separation using a Dionex Ultimate 3000 uHPLC at 220 nl/min with a gradient of 2–35% organic (0.1% formic acid in acetonitrile) over 2 hr. Peptides were analyzed using a Thermo Orbitrap Fusion mass spectrometer operating at 120,000 resolution (FWHM in MS1) with HCD sequencing at top speed (15,000 FWHM) of all peptides with a charge of 2+ or greater. The raw data were converted into *.mgf format (Mascot generic format) for searching using the Mascot 2.5.1 search engine (Matrix Science) against Rat protein sequences (Uniprot 2017). The database search results were loaded onto Scaffold Q+ Scaffold_4.4.8 (Proteome Sciences) for statistical treatment and data visualization.

## Immunofluorescence

For antibody validation, U2OS cells (parental and C9ORF72 KO) were transfected with LAMP1-YFP and LAMP1-RFP, respectively. LAMP1-YFP (Addgene plasmid #1816) and LAMP1-RFP (Addgene plasmid #1817) are gifts from Walther Mothes. At 24 hr post transfection, both cell lines were plated on glass coverslips as a mosaic and incubated for 24 hr. Cells were fixed in 4% PFA for 10 min and then washed 3 times. Cells were then blocked and permeabilized in blocking buffer (TBS, 5% BSA and 0.3% Triton X-100, pH 7.4) for 1 hr at room temperature. Coverslips were incubated face down on a 50 µl drop (on paraffin film in a moist chamber) of blocking buffer containing the primary C9ORF72 antibodies diluted at 2 µg/ml and incubated O/N at 4°C. Cells were washed 3 × 10 min and incubated with corresponding Alexa Fluor 647-conjugated secondary antibodies diluted 1:1000 in blocking buffer for 2 hr at room temperature. Cells were washed 3 × 10 min with blocking buffer and once with TBS. Coverslips were mounted on a microscopic slide using fluorescence mounting media (DAKO, Cat# S3023). HEK-293 (parental and C9ORF72 KO) were plated separately on coverslips and stained as mentioned above. HEK-293 cell lines were fixed in either 4% PFA for 10 min or in methanol (chilled at −20°C) for 10 min. Imaging was performed using a Leica SP8 laser scanning confocal microscope equipped with a 40x oil objective (NA = 1.30) and HyD detectors. Acquisition was performed using Leica Application Suite X software (version 3.1.5.16308) and analysis was done using Image J. All cell images represent a single focal plane. They were prepared for publication using Adobe Photoshop to adjust contrast, apply 1 pixel Gaussian blur and then assembled with Adobe Illustrator.

Imaging of C9ORF72 in U2OS cells was performed as above after transfecting LAMP1-RFP or GFP-Rab constructs. GFP-Rab5a, GFP-Rab7a, GFP-Rab9 and GFP-Rab11a were gifts from Mitsunori Fukuda (*Matsui et al., 2011*). Imaging was performed using a 63x oil objective (NA = 1.40) and we used LIGHTNING (Leica) to acquire images using the best setting for the resolution of the deconvolved image. Enough Z-stack were imaged to cover the whole cell depth. To evaluate co-localization between C9ORF72 and GFP-Rab5, GFP-Rab7a, GFP-Rab9 and GFP-Rab11, we collected confocal images appropriate for a co-localization study (*Dunn et al., 2011*). LIGHTNING deconvolution was applied to all images. Pearson's coefficient correlation was measured using the Coloc2

plugin from Fiji (*Schindelin et al., 2012*). We used a Student's t-test to determine if the means of the correlation coefficient between C9ORF72 and the Rabs is significantly different. \*\*\*=p value lower than 0.01.

Quantification of the percentage of C9ORF72 fluorescence signal on and outside of lysosomes was performed in three dimensions (3D) using Imaris software (Bitplane, version 9.3). Briefly, a mask was applied on the whole cell of interest by manually drawing the cell contour. The total C9ORF72 fluorescent (C9ORF72$^{total}$) signal was determined within the mask. Then the LAMP1 signal was segmented and the amount of C9ORF72 fluorescent signal found on lysosomal surfaces was calculated (C9ORF72$^{lyso}$). The percentage of C9ORF72 fluorescent signal on lysosomes was calculated as follow: C9ORF72$^{lyso}$/C9ORF72$^{total}$ X 100. Assessment of LAMP1 structures positive for C9ORF72 was performed by visual identification of C9ORF72 fluorescent signal on lysosomal surfaces. We used a Student's t-test to determine if the means of C9ORF72 fluorescent signal on lysosome or the means of LAMP1 structures positive for C9ORF72 between fed and starved condition are significantly different. n/s = p value higher than 0.05.

## Immunohistochemistry with 3,3'-diaminobenzidine (DAB) staining

Formalin-fixed, paraffin-embedded brain tissue from 3 month old C9-KO (n = 3) and C9-WT (n = 3) mice were sectioned at 6 µm in the sagittal plane and then mounted onto positively charged slides. Sections were deparaffinised at 60°C for 20 min on a heat block and then incubated in xylene (3 × 5 min). Subsequent rehydration was performed through graded ethanol washes and finally in water. For epitope unmasking, heat-induced epitope retrieval was carried out using TE9 buffer (10 mM Trizma base, 1 mM EDTA, 0.1% Tween 20, pH 9) at 110°C for 15 min. Endogenous peroxidases were quenched with 3% $H_2O_2$ in TBS for 10 min at room temperature. Slides were then blocked in 2.5% normal horse serum and 0.3% Triton X-100 in TBS for 1 hr at room temperature. GTX63448 and GTX632041 were diluted in DAKO antibody diluent (Agilent, Cat# S0809) at 1:5000 and then slides were incubated O/N at 4°C. Washes were then performed 3 × 10 min in TBS-0.1% Tween 20 (TBST) prior to secondary antibody incubation with ImmPRESS HRP horse anti-mouse IgG (Vector Labs, Cat# MP-7402), for 1 hr at room temperature. Slides were then washed 3 × 20 min in TBST. DAB staining was developed under a light microscope for between 2–10 min as per the manufacturer's instructions with the ImmPACT DAB peroxidase substrate kit (Vector Labs, Cat# SK-4105). Slides were then counterstained with Hematoxylin Solution, Gill No.1 (Sigma-Aldrich, Cat# GHS132) for 5 min at room temperature. Slides were then dehydrated with sequentially increasing graded ethanol and finally in xylene prior to coverslipping with Cytoseal 60 (Fisher Scientific, Cat# 8310–16). Micrographs were captured with a digital camera mounted on a Leica DM6000 B upright microscope with either 10X or 40X objectives using Volocity Imaging software (version 6.3.0 PerkinElmer).

## Rapid immune-isolation of lysosomes (LysoIP) and mitochondria (MitoIP)

The isolation of lysosomes and mitochondria was performed following the LysoIP protocol (*Abu-Remaileh et al., 2017*) and the MitoIP protocol (*Chen et al., 2016*), respectively, with slight modifications. Tmem192-3xHA (Addgene #102930), Tmem192-2xFlag (Addgene #102929) and 3xHA-eGFP-OMP25 (Addgene #83356) were gifts from Dr. David Sabatini. For LysoIP experiments, HEK-293 cells were transfected with either Tmem192-3xHA (HA-Lyso cells) or with TMEM192-2xFlag (Control-Lyso cells). For MitoIP, HEK-293 were transfected without (control anti-HA IP) or with the 3xHA-eGFP-OMP25 construct.

The following steps are the same for LysoIP and MitoIP: Approximately 35 million cells were used for each immunoprecipitation. Cells were quickly rinsed twice with cold PBS and scraped in 2 ml of KPBS (136 mM KCl, 10 mM $KH_2PO_4$, pH 7.25 and protease inhibitor) and centrifuged at 1000 x g for 2 min at 4°C. Pelleted cells were resuspended in 900 µl of KPBS and gently homogenized with 20 strokes in a 2 ml hand-held homogenizer. The homogenate was centrifuged at 1000 x g for 2 min at 4°C. The supernatant was collected and 10 µl (equivalent to 1,1%) was reserved and run on immunoblot as the starting material. The remaining supernatant was incubated with 150 µl of KPBS prewashed anti-HA magnetic beads (Thermo Fisher Scientific cat# 88837) on a gentle rotator shaker for 3 min. Beads were gently washed 3 times with 1 ml of KPBS using a DynaMag-2 magnet (Thermo Fisher Scientific cat# 12321D). Beads were resuspended in 1 ml of KPBS and transfer to a new tube.

KPBS was removed and beads were incubated in resuspension buffer (20 mM HEPES, 1 mM EDTA, 150 mM NaCl, 1% Triton X-100, pH 7,4 + phosphatase inhibitor). Supernatant was collected and run on an immunoblot.

### Subcellular fractionation

Cells were quickly rinsed with cold PBS and scraped in HEPES buffer (20 mM HEPES, pH 7.4) supplemented with protease inhibitors. Lysates were gently homogenized with 20 strokes in a 2 ml hand-held homogenizer and spun at 200 000 x g for 30 min at 4°C, yielding the supernatant (S) and pellet (P) fractions. Equal protein aliquots of the fractions were analyzed by quantitative immunoblot.

### Bone-marrow-derived macrophages

Bone marrow cells were isolated from femurs of around 12 weeks old mice, as described elsewhere (*Trouplin et al., 2013*). Cells were seeded in bacterial dishes and cultured in DMEM high-glucose (GE Healthcare cat# SH30081.01) containing 10% heat inactivated fetal calf serum (FBS; 080–150), 2 mM L-glutamate (Wisent cat# 609065, 100 IU penicillin and 100 µg/ml streptomycin (Wisent cat# 450201) and containing 20% L929 cell-conditioned medium as a source of M-CSF. After 7 days, cells were harvested using cold PBS, seeded, and kept in DMEM, 10% FBS and penicillin/streptomycin for 24 hr before experiment.

### Human monocyte-derived macrophages

Peripheral blood mononuclear cells (PBMCs) were isolated from whole blood using Ficoll–Paque density gradient centrifugation (GE Healthcare). CD14+ cell isolation was done using immunomagnetic bead selection according to manufacturer's instructions to achieve 95–99% purity (Miltenyi Biotec). CD14+ monocytes were cultured in RPMI cell medium supplemented with 10% FCS, 0.1% penicillin/streptomycin and 0.1% glutamine. MDMs were differentiated in vitro with 25 ng/ml M-CSF for 6 d, during which time cells they were either activated with the pro-inflammatory cytokines IFN-γ and LPS or the anti-inflammatory cytokine TGF-β, as previously described (*Healy et al., 2016*). Briefly TGF-β-treated cells received recombinant human TGF-β (20 ng/ml) on days 1 and 4. Pro-inflammatory activated cells received IFN-γ (20 ng/ml) for 1 hr followed by a 48 hr treatment with LPS (serotype 0127:B8, 100 ng/ml). The studies with human tissue followed Canadian Institutes of Health Research approved guidelines. Secondary use of de-identified tissues was carried out in accordance with the guidelines set by the McGill University Institutional Review Board (McGill University Health Centre Ethics Board and approved under protocol ANTJ1988/89). All experiments were conducted in accordance with the Helsinki Declaration with sample procured with informed consent.

## Acknowledgements

We thank Dr. Don Cleveland (UCSD) and Dr. Clothilde Lagier-Tourenne (UMass) for C9ORF72 KO mice. We thank Mitsunori Fukuda for GFP-tagged Rab constructs. Proteomics analysis was performed at the Clinical Proteomics Platform of the Research Institute of the McGill University Health Centre. Cell imaging was performed at the Neuro Microscopy Imaging Centre at McGill. We would like to thank the Advanced BioImaging Facility at McGill for their help on image analysis. This work was supported by a grant from the Motor Neurone Disease Association (UK), The ALS Association (USA) and ALS Canada and by an Arthur J Hudson Team Grant from ALS Canada/Brain Canada. CL is supported by the Ronald Peter Griggs and Tim E Noël Postdoctoral Fellowship from ALS Canada. RK is supported by a studentship from the Canada First Research Excellence Fund, awarded to McGill University for Healthy Brains for Healthy Lives. JR holds the James Hunter and Family Chair in ALS Research. PSM is a James McGill Professor and a Fellow of the Royal Society of Canada.

## Additional information

### Funding

| Funder | Grant reference number | Author |
|---|---|---|
| ALS Society of Canada | Hudson Translational Grant | Janice Robertson<br>Peter S McPherson |
| ALS Society of Canada | ALS RAP | Opher Gileadi<br>Thomas M Durcan<br>Aled M Edwards<br>Janice Robertson<br>Peter S McPherson |
| ALS Therapy Alliance | ALS RAP | Thomas M Durcan<br>Aled M Edwards<br>Janice Robertson<br>Peter S McPherson |
| Motor Neurone Disease Association | ALS RAP | Thomas M Durcan<br>Aled M Edwards<br>Janice Robertson<br>Peter S McPherson |

The funders had no role in study design, data collection and interpretation, or the decision to submit the work for publication.

### Author contributions

Carl Laflamme, Conceptualization, Data curation, Investigation, Methodology, Formal analysis, Writing—original draft, Writing—review and editing; Paul M McKeever, Data curation, Investigation, Methodology; Rahul Kumar, Data curation; Julie Schwartz, Mahshad Kolahdouzan, Carol X Chen, Zhipeng You, Faiza Benaliouad, Methodology; Opher Gileadi, Heidi M McBride, Resources; Thomas M Durcan, Luke M Healy, Janice Robertson, Resources, Methodology; Aled M Edwards, Resources, Data curation; Peter S McPherson, Conceptualization, Resources, Supervision, Funding acquisition, Writing—original draft, Writing–review and editing

### Author ORCIDs

Carl Laflamme (ID) https://orcid.org/0000-0001-5906-025X
Rahul Kumar (ID) https://orcid.org/0000-0002-1061-5194
Opher Gileadi (ID) https://orcid.org/0000-0001-6886-898X
Heidi M McBride (ID) https://orcid.org/0000-0003-4666-2280
Peter S McPherson (ID) https://orcid.org/0000-0001-7806-5662

### Ethics

Human subjects: Human monocyte derived macrophages phages were isolated from blood samples. Secondary use of de-identified tissues was carried out in accordance with the guidelines set by the McGill University Institutional Review Board (McGill University Health Centre Ethics Board and approved under protocol ANTJ1988/89). All experiments were conducted in accordance with the Helsinki Declaration with sample procured with informed consent.
Animal experimentation: Mice were bred and cared for in accordance with the guidelines of the Canadian Council on Animal Care following protocols approved by both the University of Toronto and McGill University Animal Care Committee.

### Decision letter and Author response

Decision letter https://doi.org/10.7554/eLife.48363.sa1
Author response https://doi.org/10.7554/eLife.48363.sa2

## Additional files

### Supplementary files

• Supplementary file 1. Summary of C9ORF72 antibodies and their properties.

• Supplementary file 2. Mass spectrometry data from immunoprecipitation of C9ORF72 from parental and KO cell lines.

• Transparent reporting form

### Data availability

Performance of each antibody in every application is summarized in Supplementary file 1.

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
