## [Decision Letter]

Thank you for submitting your article "Implementation of an antibody validation procedure: Application to the major ALS/FTD disease gene C9ORF72" for consideration by *eLife*. Your article has been reviewed by three peer reviewers, including Suzanne R Pfeffer as the Reviewing Editor and Reviewer #1, and the evaluation has been overseen by Michael Eisen as the Senior Editor. The following individuals involved in review of your submission have agreed to reveal their identity: Fridtjof Lund-Johansen (#2) and Dario R Alessi (Reviewer #3).

The reviewers have discussed the reviews with one another and the Reviewing Editor has drafted this decision to help you prepare a revised submission.

This paper presents a critically important evaluation of available antibodies that recognize a widely studied and important protein linked to frontotemporal dementia and ALS. The work will be of broad interest and significance to readers of *eLife* however the reviewers were slightly split in their assessments. Reviewer #2 was less convinced that the manuscript presented the best pipeline for all proteins, but felt that the authors could refer to other published pipelines and discuss the limitations mentioned in each of the reviews. Because all reviewers noted the importance of this contribution, we welcome a revised version.

Essential revisions:

It seems really important to:

1) Quantify the IF;

2) Include a mouse cell line;

3) Determine percent efficiency of IP as noted in the reviews below. It would also be valuable to consider IF/IP for the antibodies that did not blot well as some may be useful for those types of analysis.

Also, all other points should be discussed. We include for your guidance all reviews so that you can address many of the points textually.

We hope you will find these comments constructive.

Reviewer #1:

This is a timely and important TOOL contribution that highlights the importance of antibody validation in the biomedical research community. The authors characterize all commercially available antibodies that are supposed to recognize a protein important in frontotemporal dementia and ALS and show that many prior studies are incorrect.

The work will be of broad interest to readers of *eLife* after the authors address the following issues.

1) Since the authors are presenting what should be the definitive localization/characterization paper, they need to complete the IF localization with additional markers and quantitation of co-localization over a larger number of cells to report the precise localizations of the protein under study in at least two cell types. Also, membrane/cytosol fractionation is important as large cytosolic pools of proteins can interfere with proper membrane localizations and this causes many to use saponin or liquid nitrogen permeabilization to depleted cytosolic proteins before fixation (please discuss).

2) Since there are many papers with the wrong conclusions, these should be included in Table 1 (or in another table) in terms of who used which antibody, to best help workers in this field re-evaluate the literature.

Additional comment received by email: "The blot in Figure 4B appears to be underexposed if we compare it to those in Figure 1 and Figure 2—figure supplement 1. Even at this brief exposure, there are additional bands in all cell types except for HEK which were used in Figure 1 and Figure 2—figure supplement 1. We therefore need to see a blot with longer exposure time. We also need to see the full blots for murine tissues in Figure 4A. This result illustrates a key challenge with antibody validation: Cross-reactivity is often sample-dependent. Thus, a clean WB in one, or a few, cell types is not evidence for consistent specificity.”

Reviewer #2:

1) McPherson and co-authors describe a pipeline to characterize antibodies to human proteins. The article addresses an important problem in bio-medical research since a high frequency of commercially available antibodies do not perform in the applications they were specified for by the manufacturers. The pipeline involves the use of published MS data to identify a human cell line with abundant expression of the antibody target, CRISPR/Cas9 KO, identification of non-redundant commercially available antibodies to the target, western blotting of parental and KO cells, western blotting with a panel of cell lines to identify a cell type with high expression, use of antibody in IP-MS and immunofluorescence microscopy (4% PFA/1% Triton X100).

The authors also generated KO mice to test performance of the antibodies in IHC.

General assessment. The experiments are well performed and the data are convincing. However, the manuscript merely describes use of techniques that are already well-established. Moreover, the pipeline they describe has several important limitations.

2) Buying all commercially available antibodies to a given protein is hardly a "relatively inexpensive approach" to begin with. A key question is if the product specification sheets were at all useful to narrow down the number of reagents for testing. The manufacturers WBs for the two GeneTex antibodies look quite good, and the best antibody is the same as that reported here. I was unable to find data specification sheets for the catalogue numbers of most antibodies listed in Table 1. The exception was the HPA reagent, which as listed as not useful in WB. Were the WBs of all other antibodies really as convincing as those from GeneTex? This would be surprising in view of the horrible blots in Figure 2.

3) The authors initially selected HEK293 cells for KO on basis of published MS data from human cell lines. The reference data were published in 2012, and for MS this is outdated. The limitation of the reference is also evident from their results. Thus, the authors found that none of the antibodies were useful in immunofluorescence microscopy in HEK cells. Rather than sticking to their original pipeline, they now selected another cell type (U2OS) for KO on basis of a minor difference in the intensity of bands on WB. I do not find this to be particularly convincing. A better approach would be to generate a new MS dataset from the cell lines they intended to use. (Please discuss).

4) The authors also used KO cells as reference to test performance of antibodies in IP-MS. However, KO is not necessary or even particularly useful in this setting since the assay primarily reports on the ability of the antibody to enrich the intended target, and the identification is based on sequence information. Cross-reactivity is generally not assessed, and the authors did not make attempts to do so here. A more relevant reference would be shotgun MS analysis of a total cell lysate to determine enrichment in the IP as relative protein rank. This was not performed here. Please discuss (see next comment).

5) I think it would be useful to have a discussion about IP-MS. My view is that it is very difficult to assess specificity by IP-MS if one uses total cell lysates as starting material. We would see this more clearly if the authors had provided the processed MS data. IPs of native proteins from total cell lysates typically contain hundreds of proteins. Let us say you compare the list from KO and WT. Would you consider the overlap to be cross-reactivity or "CRAPOME"? Should all proteins found in IP from WT and not in KO be considered as interaction partners of the intended target? Probably not. Aled Edwards has published a paper on this topic in Nature Methods in 2015, so the authors are experienced enough to do this. My suggestion is that they revise the paper to discuss their MS data in more detail. They can certainly make claims about enrichment of the intended target, but can they really say anything about cross-reactivity?

Reviewer #3:

The paper touches upon a very significant issue of the vast majority of antibodies used in research not being properly validated. This results in confusion and lack of reproducibility in the field.

The authors describe an antibody validation workflow using the C9ORF72, one of the major ALS disease genes. They employ immunoblotting, immunoprecipitation, immunofluorescence and immunohistochemistry to validate a set of commercially available antibodies. The first step in the pipeline is immunoblot validation by screening various cell lines in order to determine expression levels of the target protein and choose the right cell line for further validation steps.

The study is very well written and very clear. Data presentation is also very clear. This study is an important addition to the literature, and I would be in favour of recommending it be published in *eLife*. There a few points that the authors may wish to consider in a revised version that are outlined below.

1) As various disease models are used employing different animal species not just humans, it would be useful to include cell lines of other species i.e. mouse at the initial immunoblotting validation, as a good antibody (e.g. anti-mouse) may be missed because the primary screen is performed in human cell lines only. It could be done especially if the knock-out animal model is available. It is the case with the C9ORF72 (knock-out mouse available) and a mouse embryonic fibroblast cell line could be easily generated to be used for validation. We often find that antibodies unfortunately do not always behave the same for mouse and human.

2) Generation of CRISPR/Cas9 knock-out in human cell lines expressing the target protein is a very effective approach to validate the antibodies both for immunoblotting and immunofluorescence. For immunofluorescence validation (Figure 4) an additional approach of generating a GFP-tag CRISPR knock-in could be used to further support the results, in which GFP signal would co-localise with the target protein signal. This also offers the advantage of knowing that you can visualise the endogenous protein before testing the IF grade antibody.

3) For testing immunoprecipitation, we always quantify percentage of protein depleted from the supernatant after immunoprecipitation. This is very important. Many antibodies that are claimed to immunoprecipitated only deplete a small percentage of the endogenous protein from the supernatant. We recently tested 167 monoclonal clones against one target and only one of these was capable at nearly quantitatively depleting the endogenous protein from the supernatant with the rest bringing down trace or small levels in comparison.

4) If the KO mouse model is available, it may be useful to include validation using wild-type and knock-out mouse embryonic fibroblasts or other cell lines before moving on to immunohistochemistry in mouse tissues (Supplementary Figure 5) (not essential).

5) For MS experiments I would recommend doing IPs just from the wild type and KO cells (no other controls needed). In addition to the bait protein you are looking for additional proteins that co-IP with bait in wild type and not KO cells.

6) MS analysis performed on the immunoprecipitates allowed to show the selected antibodies were able to enrich C9ORF72 and its interactors (SMCR8 and WDR41). However, it would help to compare the overall selectivity of the antibodies if the abundance of the target proteins versus the large number of contaminating proteins could be presented. Ideally for immunoprecipitation of endogenous proteins especially if this is going to be used in biochemical studies one needs the immunoprecipitates to be as clean as possible. Being able to catalogue the identity and number of non-specific proteins in each immunoprecipitate may be useful in selecting an antibody for such an application.

7) This article is written for researchers wishing to select which is the best available commercial antibody. However, it is very applicable to researchers raising and characterising their own antibodies-especially monoclonal antibodies where one needs to test many dozens of clones. Maybe this could be mentioned.

8) The clones that work best for immunoblot are not always the best for IP or IF analysis so we would recommend retesting all available antibodies for these applications and not just testing the antibody that works best for immunoblot analysis.

9) Maybe the authors could discuss the danger of a field relying on a small number of commercial antibodies. What happens if the supplier making these terminates the programme and no longer supplies the reagents? There is also batch to batch variation of commercial products, and I have often heard that quality of commercial antibody batches can vary. The cost of commercial antibodies is extremely high and prohibitive for many researchers. Most researchers would never have the funds to purchase dozens of commercially available antibodies for each antibody to test. I think the solution to all of this is mass spectrometry sequencing of the best set of identified commercial monoclonal antibodies. We have done this a few times and there are commercial very reliable companies who do this for a cost of around $10,000 and require <200 micrograms of monoclonal antibody (that can be purchased from supplier). This may seem expensive but once this information is available an expression construct can be made to generate large amounts of the recombinant antibody in HEK293 cells (often >10 mg/litre expression). This plasmid vector could then be given out freely to all researchers working in the field to generate widely used antibodies needed for their work. The overall cost of doing this is very low compared to the overall cost of the research being funded by foundations and agencies. Perhaps foundations and agencies could have calls put out to enable key antibodies used by research fields to be sequenced and cloned for the benefit of everyone. If the hybridoma clone for the antibody is available, the antibody sequence can be obtained very easily by PCR DNA sequencing-but for most commercial antibodies the hybridoma clone is not available.

10) Could the authors discuss where researchers who spend a lot of time characterizing antibodies could deposit this important characterisation data – i.e. make it available in a timely manner to researchers working in the field. If it just stays in the researchers lab book it is not much use for the research field. It is very unlikely that this data could be published in a conventional paper and even if it was it would be buried in likely as supplementary methods or figure section and thus very hard to locate. Some thoughts on this would be interesting.

11) Mouse ortholog gene name should be 3110043O21Rik not 31100432021.

---

## [Author Response]

This paper presents a critically important evaluation of available antibodies that recognize a widely studied and important protein linked to frontotemporal dementia and ALS. The work will be of broad interest and significance to readers of eLife however the reviewers were slightly split in their assessments. Reviewer #2 was less convinced that the manuscript presented the best pipeline for all proteins, but felt that the authors could refer to other published pipelines and discuss the limitations mentioned in each of the reviews. Because all reviewers noted the importance of this contribution, we welcome a revised version.Essential revisions:It seems really important to:1) Quantify the IF;2) Include a mouse cell line;3) Determine percent efficiency of IP as noted in the reviews below. It would also be valuable to consider IF/IP for the antibodies that did not blot well as some may be useful for those types of analysis.Also, all other points should be discussed. We include for your guidance all reviews so that you can address many of the points textually.

We responded to each of these essential revisions experimentally. The responses are outlined in detail in the individual response to reviewers. For comment 1, see response to reviewer #1, comment 1. For comment 2, see response to reviewer #3, comment 1. For comment 3, see response to reviewer #3, comment 3. For the comment regarding IF/IP for antibodies that did not blot, see response to reviewer #3, comment 8.

We hope you will find these comments constructive.Reviewer #1:This is a timely and important TOOL contribution that highlights the importance of antibody validation in the biomedical research community. The authors characterize all commercially available antibodies that are supposed to recognize a protein important in frontotemporal dementia and ALS and show that many prior studies are incorrect.The work will be of broad interest to readers of eLife after the authors address the following issues.1) Since the authors are presenting what should be the definitive localization/characterization paper, they need to complete the IF localization with additional markers and quantitation of co-localization over a larger number of cells to report the precise localizations of the protein under study in at least two cell types. Also, membrane/cytosol fractionation is important as large cytosolic pools of proteins can interfere with proper membrane localizations and this causes many to use saponin or liquid nitrogen permeabilization to depleted cytosolic proteins before fixation (please discuss).

We agree that additional experiments are required to allow us to be more definitive regarding the localization of C9ORF72. We approached this in several ways: 1) We quantified the percentage of LAMP1-positive structures in U2OS cells that were positive for C9ORF72, revealing that over 80% of lysosomes have C9ORF72 (revised Figure 6B); 2) We used immunoisolation to purify lysosomes from HEK-293 cells demonstrating that C9ORF72 is enriched on these highly enriched organelles (revised Figure 6C); 3) We compared the localization of C9ORF72 to GFP-tagged Rab5, 7, 9, 11 in U2OS cells. These experiments revealed that C9ORF72 had the highest degree of co-localization with Rab7 and 9 (revised Figure 6—figure supplement 1). Finally, in the original manuscript we demonstrated that C9ORF72 localizes to late phagosomes and phagolysosomes in RAW264.7 macrophages and this remains in the revised manuscript (revised Figure 8).

In addition, we performed subcellular fractionation studies to examine the relative membrane/cytosol pools of C9ORF72. It appears that only 20-30% of total C9ORF72 is membrane associated depending on the cell line. We tried pre-fixation permeabilization using various approaches (saponin/digitonin/liquid nitrogen treatment) but unfortunately did not achieve satisfactory results.

2) Since there are many papers with the wrong conclusions, these should be included in Table 1 (or in another table) in terms of who used which antibody, to best help workers in this field re-evaluate the literature.

All of the studies that used a C9ORF72 commercial antibody that failed validation are described in the first paragraph of the subsection “C9ORF72 localizes partially to lysosomes”. Since we did not attempt to directly reproduce these data with the high-quality antibodies, we do not feel appropriate to list them in Table 1.

Additional comment received by email: "The blot in Figure 4B appears to be underexposed if we compare it to those in Figure 1 and Figure 2—figure supplement 1. Even at this brief exposure, there are additional bands in all cell types except for HEK which were used in Figure 1 and Figure 2—figure supplement 1. We therefore need to see a blot with longer exposure time. We also need to see the full blots for murine tissues in Figure 4A. This result illustrates a key challenge with antibody validation: Cross-reactivity is often sample-dependent. Thus, a clean WB in one, or a few, cell types is not evidence for consistent specificity.”

We thank the reviewer for this comment since we did not address it in the original submission. Original Figure 4B is the validation of our knockout in U2OS cells. The reviewer is likely referring to either original Figure 4A or Supplementary Figure 5, both of which are immunoblots of cell/tissue lysates. Both of these blots were performed in a quantitative manner using the LICOR system, which uses fluorescent secondary antibodies. In this system all signals are generally weaker than when using ECL and we have noted this in the revised manuscript (Figure 4—figure supplement 3, we removed original Supplementary Figure 5). The reviewer is correct in noting that there are additional bands in several tissues and cell lysates. We also appreciate the comment regarding cross-reactivity in human versus mouse samples, a point also raised in comment 1 of reviewer #3. We thus screened all antibodies by immunoblot on mouse tissue (revised Figure 2—figure supplement 1). Note that we have added two additional antibodies that became available after the original submission. As for human samples, GTX634482 is specific and has no obvious cross-reactive bands. However, even when antibodies appear highly specific in one tissue or species, they can demonstrate cross-reactivity under different conditions. This is simply a limitation of antibody validation in general. We have noted this limitation in the subsection “Analysis by immunoprecipitation”.

Reviewer #2:1) McPherson and co-authors describe a pipeline to characterize antibodies to human proteins. The article addresses an important problem in bio-medical research since a high frequency of commercially available antibodies do not perform in the applications they were specified for by the manufacturers. The pipeline involves the use of published MS data to identify a human cell line with abundant expression of the antibody target, CRISPR/Cas9 KO, identification of non-redundant commercially available antibodies to the target, western blotting of parental and KO cells, western blotting with a panel of cell lines to identify a cell type with high expression, use of antibody in IP-MS and immunofluorescence microscopy (4% PFA/1% Triton X100).The authors also generated KO mice to test performance of the antibodies in IHC.General assessment. The experiments are well performed and the data are convincing. However, the manuscript merely describes use of techniques that are already well-established. Moreover, the pipeline they describe has several important limitations.

We feel that the use of well-established techniques is a strength as the manuscript is a guide that other laboratories can use for antibody validation. In revising the manuscript we better address some of the limitations of the pipeline although we recognize that antibody validation is a complex issue.

2) Buying all commercially available antibodies to a given protein is hardly a "relatively inexpensive approach" to begin with. A key question is if the product specification sheets were at all useful to narrow down the number of reagents for testing. The manufacturers WBs for the two GeneTex antibodies look quite good, and the best antibody is the same as that reported here. I was unable to find data specification sheets for the catalogue numbers of most antibodies listed in Table 1. The exception was the HPA reagent, which as listed as not useful in WB. Were the WBs of all other antibodies really as convincing as those from GeneTex? This would be surprising in view of the horrible blots in Figure 2.

We agree that buying all commercially available antibodies for a given protein can by expensive and using the product sheets could in principle be useful to prioritize antibodies. However, in some cases the product sheets are misleading (showing an immunoblot without clearly indicating its on overexpressed protein for example). Moreover, immunoblot is the least challenging application. For IF/IP/IHC, 11 antibodies out of the 14 used in the original submission did not perform as indicated by the companies, which in no case used knockout/knockdown controls. Thus, we tended away from using datasheets for prioritization of antibodies. That said we were able to obtain refunds for 7 antibodies out of 14 that did not perform as indicated.

We deposited our manuscript in bioRxiv in December 2018. After assessing our data, both Abcam and Genetex recently removed underperforming C9ORF72 antibodies from their catalogs, explaining why at least some datasheets were not available, and highlighting a strength of our study.

3) The authors initially selected HEK293 cells for KO on basis of published MS data from human cell lines. The reference data were published in 2012, and for MS this is outdated. The limitation of the reference is also evident from their results. Thus, the authors found that none of the antibodies were useful in immunofluorescence microscopy in HEK cells. Rather than sticking to their original pipeline, they now selected another cell type (U2OS) for KO on basis of a minor difference in the intensity of bands on WB. I do not find this to be particularly convincing. A better approach would be to generate a new MS dataset from the cell lines they intended to use. (Please discuss).

We agree with the limitations of the PaxDb dataset, but are not in a position to generate new MS datasets. This is complicated by the diversity of targets we hope the pipeline will be eventually used to address, making it difficult to generate MS datasets a priori. We are in communication with Dr. Matt Baker to gain access to deep sequencing MS datasets generate by Thermo Fisher. With continuing advances in proteomics and cell profiling selecting appropriate cell lines to begin testing should become more reliable. We have added some discussion of this issue in the second paragraph of the Discussion.

4) The authors also used KO cells as reference to test performance of antibodies in IP-MS. However, KO is not necessary or even particularly useful in this setting since the assay primarily reports on the ability of the antibody to enrich the intended target, and the identification is based on sequence information. Cross-reactivity is generally not assessed, and the authors did not make attempts to do so here. A more relevant reference would be shotgun MS analysis of a total cell lysate to determine enrichment in the IP as relative protein rank. This was not performed here. Please discuss (see next comment).

We agree that the IP-related studies deserve more discussion, which we include in the subsection “Analysis by immunoprecipitation”. While shotgun MS of the starting lysates for the IPs would aid in understanding enrichment, comment 3 of reviewer #3 also suggested measuring depletion of C9ORF72 from the unbound material of the IP as a measure of enrichment. We performed this experiment, which is presented in Figure 3—figure supplement 1 of the revised manuscript. We agree that using knockout lines in IP screens is not useful. We in fact only used knockout lines with the one antibody (GTX632041) that provided the seaming strongest enrichment of C9ORF72 (Figure 3 of the original submission). As described in response to comment 5 of reviewer #2, we have expanded these studies and this helped us in understanding issues related to specificity versus cross-reactivity.

5) I think it would be useful to have a discussion about IP-MS. My view is that it is very difficult to assess specificity by IP-MS if one uses total cell lysates as starting material. We would see this more clearly if the authors had provided the processed MS data. IPs of native proteins from total cell lysates typically contain hundreds of proteins. Let us say you compare the list from KO and WT. Would you consider the overlap to be cross-reactivity or "CRAPOME"? Should all proteins found in IP from WT and not in KO be considered as interaction partners of the intended target? Probably not. Aled Edwards has published a paper on this topic in Nature Methods in 2015, so the authors are experienced enough to do this. My suggestion is that they revise the paper to discuss their MS data in more detail. They can certainly make claims about enrichment of the intended target, but can they really say anything about cross-reactivity?

As indicated in response to comment 4, we have expanded our discussion regarding IP-MS, particularly as it relates to understanding cross-reactivity and enrichment (subsection “Analysis by immunoprecipitation”). For cross-reactivity, Dr. Lund-Johansen is correct regarding the difficulty in understanding cross-reactivity versus “CRAPOME”. In Figure 3 of the revised manuscript, we have expanded the IP-MS studies and have included additional information in the figure, which provides key talking points for the Discussion. SMCR8 and WD41 (along with C9ORF72) are both found with robust peptide counts in the parental lysates and are absent from the knockout lysates. A priori, one would take this to indicate that SMCR8 and WD41 are binding partners of C9ORF72 or are otherwise functionally linked (change in levels or functional form) with loss of C9ORF72, and from previous studies we know both of these things to be true. We also see clathrin heavy chain (CLH1) with high peptide counts, but in both the parental (85, 72, 58 peptides) and knockout (75, 51, 53 peptides) from 3 independent IPs. Is this cross-reactivity or another form of non-specific interaction, very difficult to know? Finally, we identified MMS19 (a 113 kDa protein component of the iron-sulfur protein assembly complex) with 5, 1 and 2 peptides in IPs from the parental cells and zero in the IP from the knockout cells. This suggests a novel binding partner. Beyond that, all other identifications were with small numbers of peptides and were also identified in the knockout lines. This suggests non-specific interactions other than cross-reactivity but again this is difficult to assess. We now include all of the MS data in Supplementary file 1 of the revised manuscript.

Reviewer #3:The paper touches upon a very significant issue of the vast majority of antibodies used in research not being properly validated. This results in confusion and lack of reproducibility in the field.The authors describe an antibody validation workflow using the C9ORF72, one of the major ALS disease genes. They employ immunoblotting, immunoprecipitation, immunofluorescence and immunohistochemistry to validate a set of commercially available antibodies. The first step in the pipeline is immunoblot validation by screening various cell lines in order to determine expression levels of the target protein and choose the right cell line for further validation steps.The study is very well written and very clear. Data presentation is also very clear. This study is an important addition to the literature, and I would be in favour of recommending it be published in eLife. There a few points that the authors may wish to consider in a revised version that are outlined below.1) As various disease models are used employing different animal species not just humans, it would be useful to include cell lines of other species i.e. mouse at the initial immunoblotting validation, as a good antibody (e.g. anti-mouse) may be missed because the primary screen is performed in human cell lines only. It could be done especially if the knock-out animal model is available. It is the case with the C9ORF72 (knock-out mouse available) and a mouse embryonic fibroblast cell line could be easily generated to be used for validation. We often find that antibodies unfortunately do not always behave the same for mouse and human.

This is an important point that we have addressed experimentally, especially since high-quality antibodies for murine C9ORF72 would be useful for the ALS community as C9ORF72 is being studied in ALS-mouse models. We thus screened all antibodies by immunoblot on lysates of wild-type and knockout mouse brain (Figure 2—figure supplement 1 of the revised manuscript). Note that we have added two additional antibodies that became available after the original submission. We found that the best antibodies for immunoblot of human samples, GTX634482, ab221137 and MRC-478D perform as well on murine samples. We were surprised to observe that some antibodies that did not detect C9ORF72 specifically in human lysates could identify C9ORF72 in mouse samples. These antibodies are GTX119776, ab227555 and sc-138763.

2) Generation of CRISPR/Cas9 knock-out in human cell lines expressing the target protein is a very effective approach to validate the antibodies both for immunoblotting and immunofluorescence. For immunofluorescence validation (Figure 4) an additional approach of generating a GFP-tag CRISPR knock-in could be used to further support the results, in which GFP signal would co-localise with the target protein signal. This also offers the advantage of knowing that you can visualise the endogenous protein before testing the IF grade antibody.

We agree that this could be a very effective way to ensure visualization of the protein and determine if the antibody can detect the protein at endogenous levels. We have mentioned this idea in the subsection “Analysis by immunofluorescence”. In fact we originally knocked in a 2X Flag-tag to C9ORF72 in HEK-293. Surprisingly, IF with Flag antibodies revealed the same fluorescent intensity in both parental and knockin cell lines.

3) For testing immunoprecipitation, we always quantify percentage of protein depleted from the supernatant after immunoprecipitation. This is very important. Many antibodies that are claimed to immunoprecipitated only deplete a small percentage of the endogenous protein from the supernatant. We recently tested 167 monoclonal clones against one target and only one of these was capable at nearly quantitatively depleting the endogenous protein from the supernatant with the rest bringing down trace or small levels in comparison.

We thank the reviewer for this comment. We have redone the IP screens and analyzed the unbound fractions. In Figure 3—figure supplement 1 we show quantitative immunoblots used to quantify the amount of C9ORF72 remaining in the unbound fraction compared to starting material. We found that only GTX632041 could significantly deplete C9ORF72 from the unbound fraction.

4) If the KO mouse model is available, it may be useful to include validation using wild-type and knock-out mouse embryonic fibroblasts or other cell lines before moving on to immunohistochemistry in mouse tissues (Supplementary Figure 5) (not essential).

This is a very reasonable point. In this case the authors include an expert (Dr. Robertson) for whom the experiments of immunohistochemistry are rapid and straight-forward.

5) For MS experiments I would recommend doing IPs just from the wild type and KO cells (no other controls needed). In addition to the bait protein you are looking for additional proteins that co-IP with bait in wild type and not KO cells.

As indicated in response to reviewer #2, comment 5, we have expanded our discussion regarding IP-MS (subsection “Analysis by immunoprecipitation”). We have also performed new IP-MS experiments (revised Figure 3) in which we have used parental versus knockout cells with no additional controls. SMCR8 and WD41 (along with C9ORF72) are both found with robust peptide counts in the parental lysates and are absent from the knockout lysates, consistent with previous studies demonstrating that SMCR8 and WD41 are binding partners of C9ORF72. We also see clathrin heavy chain (CLH1) with high peptide counts, but in both the parental (85, 72, 58 peptides) and knockout (75, 51, 53 peptides) IPs, suggesting that it is either cross-reactive with the antibody or reflects another form of non-specific IP. Finally, we identified MMS19, a 113 kDa protein component of iron-sulfer protein assembly complex with 5, 1, 2 peptides in the parental IPs and zero in the IPs from the knockout cells, suggesting a novel binding partner.

6) MS analysis performed on the immunoprecipitates allowed to show the selected antibodies were able to enrich C9ORF72 and its interactors (SMCR8 and WDR41). However, it would help to compare the overall selectivity of the antibodies if the abundance of the target proteins versus the large number of contaminating proteins could be presented. Ideally for immunoprecipitation of endogenous proteins especially if this is going to be used in biochemical studies one needs the immunoprecipitates to be as clean as possible. Being able to catalogue the identity and number of non-specific proteins in each immunoprecipitate may be useful in selecting an antibody for such an application.

As indicated in response to reviewer #2, comments 4 and 5 we have expanded our discussion regarding issues of non-specificity/cross-reactivity in IPs analyzed MS. For cross-reactivity, we feel Dr. Lund-Johansen is correct regarding the difficulty in understanding cross-reactivity versus “CRAPOME”, i.e., other forms of non-specificity. In Figure 3 of the revised manuscript, we have expanded the IP-MS studies and have included additional information in the figure, which provides key talking points for the Discussion. SMCR8 and WD41 (along with C9ORF72) are both found with robust peptide counts in the parental lysates and are absent from the knockout lysates. A priori, one would take this to indicate that SMCR8 and WD41 are binding partners of C9ORF72 or are otherwise functionally linked (change in levels or functional form) with loss of C9ORF72, and from previous studies we know both of these things to be true. In revised Figure 3 we also present data on clathrin heavy chain and MMS19 as described in response to comment 5. Finally, we present the full list of peptides found in IP from parental and knockout lines in Supplementary file 2.

7) This article is written for researchers wishing to select which is the best available commercial antibody. However, it is very applicable to researchers raising and characterising their own antibodies-especially monoclonal antibodies where one needs to test many dozens of clones. Maybe this could be mentioned.

We have now mentioned this in the first paragraph of the Discussion.

8) The clones that work best for immunoblot are not always the best for IP or IF analysis so we would recommend retesting all available antibodies for these applications and not just testing the antibody that works best for immunoblot analysis.

We did test all antibodies in IP and IF in the original manuscript (original Figures 3 and 4 and Supplementary Figure 4) and this data remains in Figures 3 and 4 of the revised manuscript. The only application where we were selective was IHC. We moved the IF screening data from supplementary data to the main figure to highlight the screening process.

9) Maybe the authors could discuss the danger of a field relying on a small number of commercial antibodies. What happens if the supplier making these terminates the programme and no longer supplies the reagents? There is also batch to batch variation of commercial products, and I have often heard that quality of commercial antibody batches can vary. The cost of commercial antibodies is extremely high and prohibitive for many researchers. Most researchers would never have the funds to purchase dozens of commercially available antibodies for each antibody to test. I think the solution to all of this is mass spectrometry sequencing of the best set of identified commercial monoclonal antibodies. We have done this a few times and there are commercial very reliable companies who do this for a cost of around $10,000 and require <200 micrograms of monoclonal antibody (that can be purchased from supplier). This may seem expensive but once this information is available an expression construct can be made to generate large amounts of the recombinant antibody in HEK293 cells (often >10 mg/litre expression). This plasmid vector could then be given out freely to all researchers working in the field to generate widely used antibodies needed for their work. The overall cost of doing this is very low compared to the overall cost of the research being funded by foundations and agencies. Perhaps foundations and agencies could have calls put out to enable key antibodies used by research fields to be sequenced and cloned for the benefit of everyone. If the hybridoma clone for the antibody is available, the antibody sequence can be obtained very easily by PCR DNA sequencing-but for most commercial antibodies the hybridoma clone is not available.

This is an intriguing idea that can reduce variability issues. Thermo Fisher is generating Affinity antibodies that are recombinant that can also deal with this issue.

10) Could the authors discuss where researchers who spend a lot of time characterizing antibodies could deposit this important characterisation data-i.e. make it available in a timely manner to researchers working in the field. If it just stays in the researchers lab book it is not much use for the research field. It is very unlikely that this data could be published in a conventional paper and even if it was it would be buried in likely as supplementary methods or figure section and thus very hard to locate. Some thoughts on this would be interesting.

The reviewer raises an excellent point, to which there is no community-accepted solution at present. However, we are planning to promote data as follows:

1) Through disease foundation networks. This project is funded by three ALS charities – the ALS Association (USA), the Motor Neurone Disease Association (UK), and the ALS Society of Canada under the auspices of a program called the ALS-Reproducible Antibody Platform (ALS-RAP). While not a general approach, for ALS, we believe that these associations provide an excellent communication vehicle to the target community. In addition, the platform has a website (https://als-rap.org/) that describes all the ALS targets being studied. The C9ORF72 validated antibodies section will be updated immediately.

2) Through commercial antibody providers. Most antibodies are procured through commercial vendors. Providing high-quality characterization data to antibody vendors is the most direct way to reach scientists. One might imagine some form of “seal” for antibodies that have been characterized to rigourous community standards – ideally by an unbiased third party. Some vendors (Abcam, Genetex and Thermo) are already removing antibodies from their catalogues when provided with conclusive evidence of non-selectivity. Genetex plans to link our manuscript to their (high-quality) C9ORF72 antibodies on their website.

3) Through disease-specific forums. We have already been disseminating the conclusions of our study our validation data on different ALS forums.

4) Pre-print servers. Although the results from a proper characterization may not merit publication in an academic journal, there is an argument to collate and place the data in a manuscript on the biorXiv. This will be searchable and will contain the data, but the manuscript will not have to pass the “novelty” bar that many journals adhere to.

5) Creating an organization to carry out unbiased antibody characterization. This long-term solution may prove the most impactful. In essence, the community would create an organization that characterizes antibodies according to a widely-accepted standard operating procedure. This non-profit hub would characterize commercial antibodies (for a fee, and thus be sustainable) and place the results into the public domain.

11) Mouse ortholog gene name should be 3110043O21Rik not 31100432021.

We thank the reviewer for pointing this out. It has been corrected in the revised manuscript.